# ReaForest: Fostering Generative Video Reasoning for Spatial Planning

**Kun Ouyang** [1]  **Yuanxin Liu** [1]  **Xinhao Li** [2]  **Linli Yao** [1]  **Xiangyu Zeng** [2]  **Haoning Wu** [3]  **Hao Zhou** [4]
**Fandong Meng** [4]  **Jie Zhou** [4]  **Xu Sun** [1]

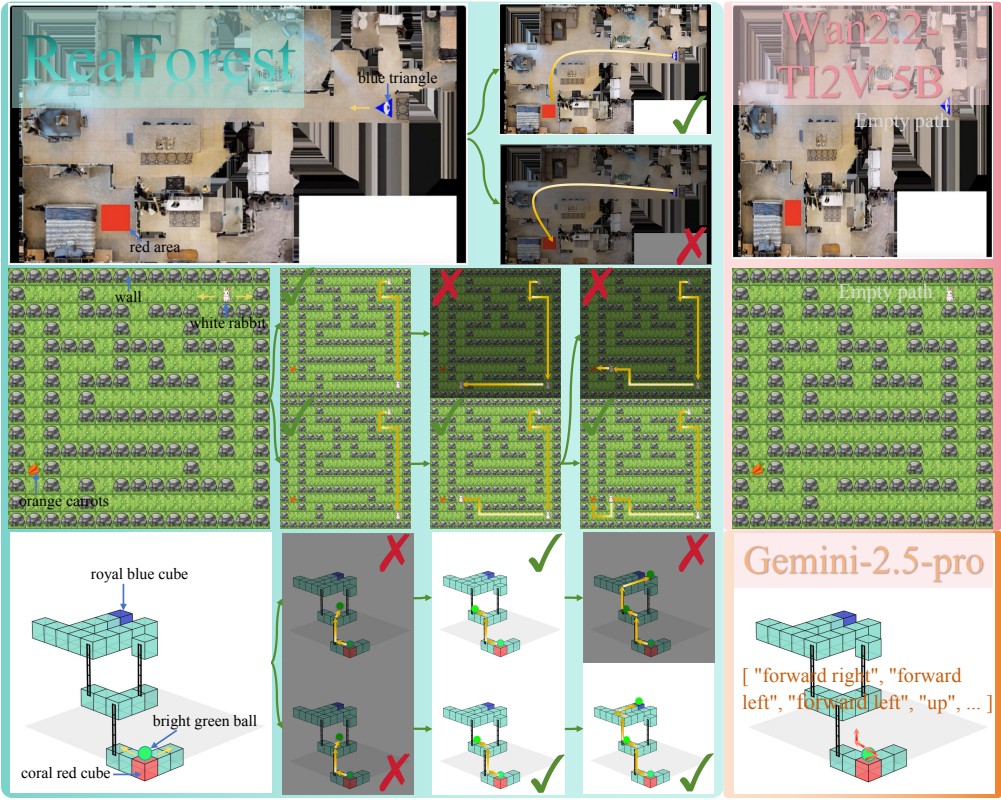

*Figure 1.* Qualitative illustration of **ReaForest**, Wan2.2-TI2V-5B, and Gemini-2.5-Pro on several spatial planning tasks.

## Abstract

Verbal logic and visual mental simulation are two essential components of human intelligence. Modern Large Language Models (LLMs) have demonstrated strong verbal reasoning capabilities through textual Chain-of-Thought (CoT) reasoning. In contrast, current Video Generation Models (VGMs) struggle with visual reasoning tasks such as spatial planning. We attribute this limitation to two fundamental gaps: (i) VGMs are

predominantly trained on general-purpose video corpora emphasizing perceptual fidelity over visual reasoning, leaving reasoning abilities underdeveloped; (ii) most VGMs generate videos in a single pass without mechanisms to explore alternative reasoning trajectories and to revise intermediate errors. Motivated by these limitations, we introduce **ReaForest**, a framework that fosters the reasoning capacity of VGMs in spatial planning through both training-time activation and inference-time scaling. ReaForest comprises three key components: **(1)** ReaGen-27k, a dataset covering diverse spatial planning tasks that require multi-step reasoning, which activates basic reasoning capabilities of VGMs for spatial planning; **(2)** Reflective Entropy-Aware Test-Time Scaling (ReaTTS), an inference framework that evolves

---

[1]State Key Laboratory for Multimedia Information Processing, School of Computer Science, Peking University [2]Nanjing University [3]Nanyang Technological University [4]WeChat AI, Tencent Inc., China. Correspondence to: Xu Sun <xusun@pku.edu.cn>.

*Proceedings of the $43^{rd}$ International Conference on Machine Learning*, Seoul, South Korea. PMLR 306, 2026. Copyright 2026 by the author(s).

multiple reasoning branches while enabling failure recovery; (3) Hierarchical constraint verification, which provides actionable feedback for ReaTTS based on decomposed constraints. Extensive experiments demonstrate that ReaForest substantially surpasses advanced textual reasoning models (*e.g.,* Gemini-2.5-Pro) and video generation models (*e.g.,* Sora-2). ReaForest exhibits emergent properties including self-correction, parallel thinking, and scalable reasoning, advancing VGMs toward human-like visual mental simulation.

# 1. Introduction

Chain-of-Thought (CoT) reasoning (Wei et al., 2022) has emerged as a powerful paradigm for enabling large language models (LLMs) (Jaech et al., 2024; Guo et al., 2025a) and multimodal large language models (MLLMs) (Comanici et al., 2025; Team et al., 2025) to solve complex reasoning tasks. However, existing CoT methods (Feng et al., 2025; Li et al., 2025) are predominantly text-centric, making them ill-suited for tracking evolving visual states across extended sequences. As a result, their effectiveness remains limited on long-horizon, vision-centric reasoning tasks (Ivanitskiy et al., 2023; Wu et al., 2024; Liu et al., 2025b).

Motivated by this limitation, Wiedemer et al. introduced the concept of Chain-of-Frame (CoF) reasoning, where the reasoning process unfolds directly within the visual domain. They observe that video generation models (VGMs) like Veo-3 (DeepMind, 2025), trained on web-scale video data, exhibit emerging CoF reasoning capabilities by generating sequential video frames that serve as visual reasoning steps. This finding highlights the promise of CoF reasoning to tackle vision-centric tasks that require long-horizon planning and multi-step reasoning, such as maze navigation.

Nevertheless, recent systematic evaluations (Cai et al., 2025; Yang et al., 2025; Luo et al., 2025) reveal that general-purpose VGMs (Wan et al., 2025; Zheng et al., 2024) still fall far short of delivering robust CoF reasoning, also referred to as generative video reasoning. They even struggle with basic spatial planning tasks like maze navigation and Sokoban puzzles. We attribute this gap to two fundamental factors. **First**, existing VGMs are primarily trained on general-purpose video corpora emphasizing perceptual fidelity over visual reasoning, resulting in limited reasoning abilities. **Second**, most VGMs generate an entire video in a single pass, lacking mechanisms to explore alternative reasoning trajectories and to revise erroneous intermediate steps. These shortcomings motivate our central research question: **How can we foster generative video reasoning capabilities in VGMs for fundamental spatial planning?**

Addressing this question necessitates tackling three key challenges. The **first challenge** is constructing a dataset that effectively activates the latent reasoning capacity of VGMs. To this end, we introduce **ReaGen-27k**, a dataset encompassing representative spatial planning tasks that require multi-step visual reasoning. Unlike standard text-image-to-video datasets where task descriptions are specified solely in text while the image defines only the initial pixel space, we propose a text-image alignment mechanism that explicitly grounds textual reasoning signals into the visual pixel space.

The **second challenge** is designing an efficient test-time scaling framework rather than single-pass generation to amplify reasoning capabilities at inference. We address this by proposing reflective entropy-aware test-time scaling (**ReaTTS**), a framework that evolves multiple reasoning branches via iterative next-clip generation based on two key mechanisms: *(a) Reflective Rectification*, which diagnoses and corrects systematic failures by modifying generation conditions instead of irrevocably pruning all branches, enabling recovery from dead ends. *(b) Entropy-Aware Budget Reallocation*, which adaptively balances exploration and exploitation via dynamically controlling the growth budget of each branch from an information-theoretic perspective.

The **third challenge** is providing comprehensive verification and actionable feedback to guide test-time scaling. In light of this, we design hierarchical constraint verification that decomposes reasoning verification into *hard*, *soft*, and *terminal* constraints. This protocol evaluates these constraints sequentially, providing step-wise feedback to guide pruning, failure diagnosis, and budget allocation in ReaTTS.

ReaForest integrates these three components to systematically foster generative video reasoning in VGMs for spatial planning. To validate its effectiveness, we conduct extensive experiment on two generative video reasoning benchmarks: VRBench (Yang et al., 2025) and MMGR (Cai et al., 2025), including six spatial planning tasks. Empirical results demonstrate that ReaForest substantially improves the reasoning capabilities of Wan2.2-TI2V-5B, achieving 88.4% on VRBench and 56.7% on MMGR in primary metrics. Qualitative comparisons in *Figure 1* further exhibit ReaForest's superiority: while Gemini-2.5-Pro and Wan2.2-TI2V-5B fail to generate coherent reasoning trajectories, ReaForest successfully reaches optimal solutions. These findings establish VGMs as effective visual reasoners for spatial planning, marking a meaningful step toward visual mental simulation. Our main contributions are:

- We introduce **ReaGen-27k**, a dataset with text-image alignment mechanism that activates foundational reasoning capabilities of VGMs for basic spatial planning.
- We propose **ReaTTS**, a test-time scaling framework featuring reflective rectification and entropy-aware budget reallocation, accompanied by hierarchical constraint veri-

fication, to amplify reasoning capacity at inference time.
- Extensive evaluations demonstrate that **ReaForest** achieves substantial improvements on several spatial planning tasks, while exhibiting emergent properties such as self-correction, parallel thinking, scalable reasoning.

## 2. Methodologies

In this section, we introduce **ReaForest**, comprising three key components as illustrated in *Figure 2*, to foster generative video reasoning for spatial planning.

### 2.1. Generative Video Reasoning Activation

To activate fundamental reasoning abilities, we construct a **Gen**erative Video **Rea**soning dataset, named **ReaGen-27k**.

**Data Generation.** We employ a custom virtual engine (Tong et al., 2025) to automatically generate raw data covering several vision-centric long-horizon reasoning tasks, including maze navigation (regular, irregular, and 3D variants), Sokoban puzzles, and trapfield traversal. These tasks share the core requirement of spatial planning and multi-step reasoning, yet exhibit diverse spatial structures and decision patterns, ranging from pathfinding in continuous spaces to discrete object manipulation and obstacle-aware navigation, thereby covering a wide spectrum of visual reasoning challenges. For each task, we generate $4,320$-$7,200$ instances spanning varying difficulty levels and layout configurations, $27,360$ in total. Each instance consists of: (i) a textual task description $T$, (ii) an initial conditioning image $I$, (iii) a target video $V$, and (iv) meta-information containing object descriptions, spatial layouts, and ground-truth trajectories.

**Text-Image Alignment.** To alleviate the burden of cross-modal semantic alignment (Fang et al., 2025) between textual task description and initial conditioning image, we explicitly inject reasoning signals of linguistic description into the visual pixel space by instantiating two complementary visual primitives: (1) *Object references*, which ground object descriptions to specific image regions based on annotated spatial layouts, reducing object ambiguity and enabling object-centric reasoning within native visual representations; and (2) *Reasoning hints*, which use directional arrows to indicate feasible motion directions without prescribing explicit trajectories. This alignment process yields the final dataset ReaGen-27k. We provide potential impact and applications of this mechanism in Appendix A.1.

**Supervised Fine-tuning.** We fine-tune the diffusion model Wan2.2-TI2V-5B (Wan et al., 2025) on ReaGen-27k, establishing the foundation for subsequent test-time scaling. The dataset statistics and other details are in Appendix A.2.

### 2.2. Reflective Entropy-Aware Test-Time Scaling

To amplify generative video reasoning capabilities at inference time, we propose **Reflective Entropy-Aware Test-Time Scaling** (ReaTTS) framework (Algorithm 1).

**Preliminaries**. Given a textual task description $T$, initial conditioning image $I$, generation budget $G$, and maximum reasoning steps $S$, our objective is to synthesize an optimal video trajectory $V^\star$. We formulate this as a search over a reasoning forest $\mathcal{F}$, where each branch $b_i^s$ represents a reasoning trajectory–a sequence of clips $[c_i^1, c_i^2, \ldots, c_i^s]$ generated up to step $s$. Each branch is assigned a growth budget $g_i^s \in \mathbb{Z}^+$ indicating the number of candidate clips to generate at the next step. Let $\mathcal{B}_s$ denote the set of active branches at step $s$.

---

**Algorithm 1** Reflective Entropy-Aware Test-Time Scaling

**Require:** Text $T$, image $I$, budget $G$, max steps $S$
**Ensure:** Best video trajectory $V^\star$
1: Initialize current step $s \leftarrow 0$
2: Initialize forest with branches $\mathcal{B}_0 \leftarrow \{b_1^0, \cdots, b_G^0\}$
3: Initialize growth budget $g_i^0 \leftarrow 1$ for $i = 1, \ldots, G$
4: **for** $s = 1$ to $S$ **do**
5:    **for** $b_i^{s-1} \in \mathcal{B}_{s-1}$ **do**
6:       Extend $b_i^{s-1}$ via generating $g_i^{s-1}$ candidate clips
7:       Update $\mathcal{B}_s \leftarrow \{b_1^s, \cdots, b_G^s\}$
8:    **end for**
9:    Prune branch $\mathcal{B}_s \leftarrow \mathcal{B}_s \setminus \{b_i^s \in \mathcal{B}_s \mid b_i^s$ violates hard constraints$\}$
10:   Obtain ratings $\mathcal{R}_s = \{r_1^s, \cdots, r_{|\mathcal{B}_s|}^s\}$
11:   Terminate branch $(\mathcal{B}_S, \mathcal{B}_s) \leftarrow (\mathcal{B}_S \cup \mathcal{T}_s, \mathcal{B}_s \setminus \mathcal{T}_s)$, where $\mathcal{T}_s = \{b_i^s \in \mathcal{B}_s \mid b_i^s$ reaches terminal constraints$\}$
12:   **if** $\mathcal{B}_s = \emptyset$ **then**
13:      Rectify $T$ and regenerate clips from step $s - 1$
14:   **else**
15:      Compute $p_i^s \propto \exp(r_i^s / \tau_s)$ (Eq. (3a))
16:      Update budget $g_i^s \leftarrow \max(1, \lfloor G \cdot p_i^s \rfloor)$
17:   **end if**
18: **end for**
19: Select $b^\star \leftarrow \arg\max_{b_i \in \mathcal{B}_S} \frac{r_i^s}{s}$
20: **return** $V^\star \leftarrow \text{Concat}(b^\star)$

---

ReaTTS, featuring Reflective Rectification and Entropy-Aware Budget Reallocation, evolves multiple reasoning branches in a forest-like structure through forest initialization, forest growth, and branch selection.

**Forest Initialization** (s=0). Each branch $b_i^0$ is rooted at a perturbed conditioning image $I_i$, obtained by varying font sizes, colors, and positions of visual primitives in text-image alignment. All branches start with unit growth budget.

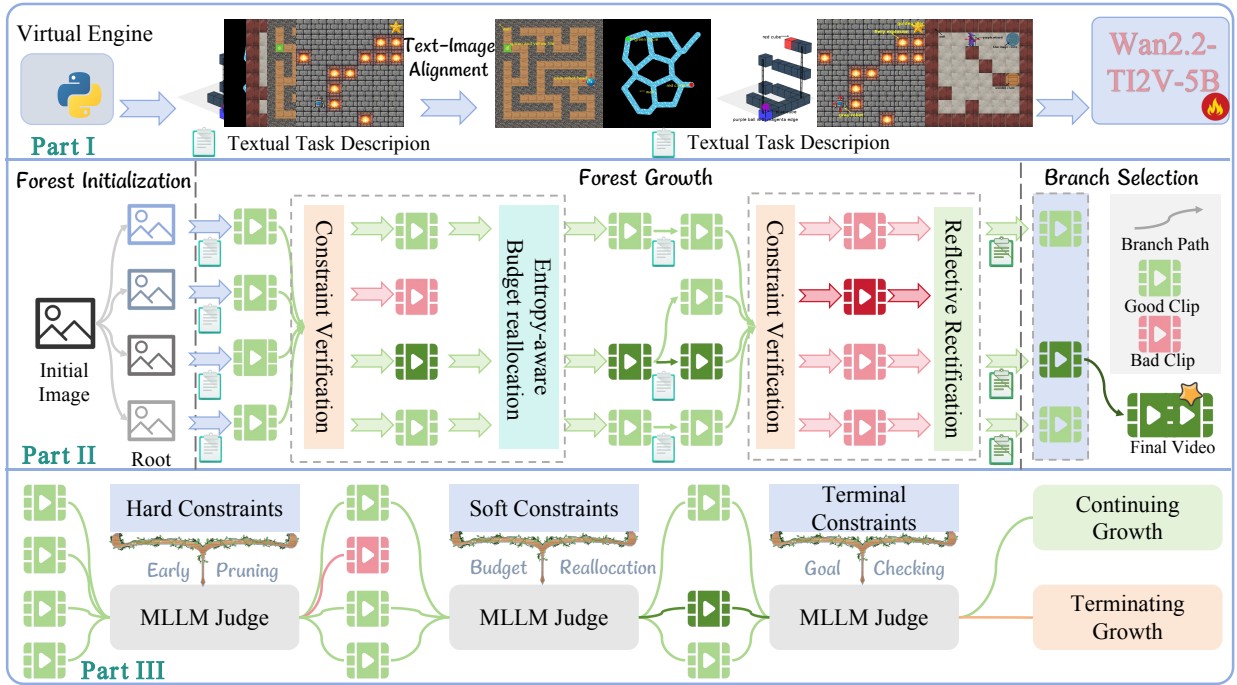

*Figure 2.* Overview of ReaForest. **Part I**: Generative Video Reasoning Activation. **Part II**: Reflective Entropy-Aware Test-Time Scaling. **Part III**: Hierarchical Constraint Verification.

**Forest Growth** (s=1,...,S). At each reasoning step $s$, we execute a three-phase growth cycle as follows.

- *Phase 1: Next Clip Generation.* For each active branch $b_i^{s-1}$ with growth budget $g_i^{s-1} \geq 1$, we generate $g_i^{s-1}$ candidate clips by conditioning on: (a) the root image $I_i$ ($s = 1$) or the final frame ($s > 1$) of the last clip $c_i^{s-1}$, (b) the task description $T$, and (c) a distinct random seed. When $g_i^{s-1} = 1$, a single clip $c_i^s$ deterministically extends the branch: $b_i^s = b_i^{s-1} \oplus c_i^s$. When $g_i^{s-1} > 1$, multiple candidates spawn separate branches that share the same prefix $b_i^{s-1}$ but diverge at step $s$.

- *Phase 2: Constraint Verification.* Each branch $b_i^s$ is evaluated by an MLLM-based verifier (§ 2.3), which identifies violated constraints and returns an incremental rating $\Delta r_i^s$ measuring reasoning quality. Branches violating *hard constraints* (§ 2.3) are immediately pruned:

$$\mathcal{B}_s \leftarrow \mathcal{B}_s \setminus \{b_i^s \in \mathcal{B}_s \mid b_i^s \text{ violates hard constraints}\}. \quad (1)$$

This early pruning prevents resource waste on fundamentally invalid trajectories. The accumulated rating is computed as $r_i^s = r_i^{s-1} + \Delta r_i^s$, with $r_i^0 = 0$. Meanwhile, branches satisfying *terminal constraints* (§ 2.3. $\mathcal{T}_{\text{sat}}$ = true) are marked as complete and directly moved to final candidates set $\mathcal{B}_S$ for branch selection.

- *Phase 3:* We handle two cases based on whether active branches remain: **(1) Reflective Rectification** ($\mathcal{B}_s = \emptyset$). When all branches are pruned, we trigger reflective recti-

fication rather than terminating the search. Specifically, we use violations identified by the verifier (*e.g.,* "collision with wall") together with the original task description $T$ to prompt an LLM, which produces augmented prompts emphasizing constraint satisfaction. These prompts are then used to regenerate the video clip from step $s-1$. This process treats failures as diagnostic signals, enabling recovery from systematic errors. **(2) Entropy-Aware Budget Reallocation** ($\mathcal{B}_s \neq \emptyset$). When active branches remain, we reallocate growth budgets across branches. Unlike hard pruning or fixed top-$k$ selection, which often causes premature diversity collapse (Xie et al., 2024; Hao et al., 2024), we adopt an entropy-aware strategy grounded in information-theoretic principles (Appendix D) to balance exploration and exploitation. Concretely, accumulated branch ratings $\mathcal{R}_s = \{r_1^s, \ldots, r_{|\mathcal{B}_s|}^s\}$ are converted into a probability distribution using a temperature-scaled softmax:

$$p_i^s = \frac{\exp\left(r_i^s / \tau_s\right)}{\sum_{k=1}^{|\mathcal{B}_s|} \exp\left(r_k^s / \tau_s\right)}. \quad (2)$$

The temperature $\tau_s$ controls allocation sharpness, where higher values encourage exploration and lower values favor exploitation. To enable a smooth transition as reasoning progresses, $\tau_s$ is annealed by step $s$ and modulated

by the normalized forest entropy $\tilde{H}_{s-1}$ (with $\tilde{H}_0 = 1$):

$$\tau_s = \tau_0 \cdot \frac{1 + \tilde{H}_{s-1}}{\exp(s/S)}, \tag{3a}$$

$$\tilde{H}_{s-1} = -\frac{1}{\log |\mathcal{B}_{s-1}|} \sum_{k=1}^{|\mathcal{B}_{s-1}|} p_k^{s-1} \log p_k^{s-1}. \tag{3b}$$

This design provides both temporal annealing and state-dependent adaptivity: high entropy sustains exploration among competing branches, while low entropy concentrates computation when a dominant branch emerges. Detailed theoretical analysis is provided in Appendix D.2. Finally, each branch $b_i^s$ receives the next-step budget:

$$g_i^{s+1} = \max\left(1, \lfloor G \cdot p_i^s \rfloor\right), \tag{4}$$

with fractional remainders redistributed to ensure $\sum_{k=1}^{|\mathcal{B}_s|} g_k^{s+1} = G$.

Growth terminates when maximum step $S$ is reached or all branches satisfy terminal constraints (§ 2.3).

**Branch Selection**. We select the best branch $b^\star$ with maximum average rating:

$$b^\star = \arg\max_{b_i \in \mathcal{B}_S} \frac{r_i^s}{s} \tag{5}$$

Finally, we construct the output video $V^\star$ by temporally concatenating all clips in the selected branch $b^\star$.

### 2.3. Hierarchical Constraint Verification

**Hierarchical Constraints.** To ensure comprehensive verification and provide reliable feedback for ReaTTS, we design a hierarchical constraint verification protocol that guides an MLLM as verifier. Unlike prior approaches (Fei et al., 2024; He et al., 2024), which assign heuristic scores to entire videos, our protocol reformulates verification as explicit constraint satisfaction over temporally ordered clips based on three decomposed constraints to comprehensively evaluate reasoning quality:

- *Hard constraints* define rule-compliance conditions that must never be violated (*e.g.,* colliding with obstacles or entering forbidden regions).
- *Soft constraints* capture desirable but non-mandatory properties of rational reasoning, such as motion smoothness, object shape consistency, and background stability.
- *Terminal constraints* verify goal satisfaction of specific tasks at clip endpoints.

**Verification Formulation.** The MLLM-based verifier $\mathcal{V}$ evaluates clips sequentially following the generation order and returns a structured assessment based on the predefined constraints:

$$\mathcal{V}(c_i^s) = (\mathcal{H}_{\text{viol}}, \Delta r_i^s, \mathcal{T}_{\text{sat}}), \tag{6}$$

where $\mathcal{H}_{\text{viol}}$ denotes the set of violated hard constraints, $\Delta r_i^s \in \mathbb{Z}$ quantifies soft constraint compliance (higher is better), and $\mathcal{T}_{\text{sat}} \in \{\text{true}, \text{false}\}$ indicates terminal goal achievement. This enables ReaTTS to perform early pruning (via $\mathcal{H}_{\text{viol}}$), budget reallocation (via $\Delta r_i^s$), and goal checking (via $\mathcal{T}_{\text{sat}}$). We provide more details such as constraint examples in Appendix B.

## 3. Experiment

### 3.1. Experiment Setup

**Benchmarks & Metrics**. We evaluate on two representative generative video reasoning benchmarks: VRBench (Yang et al., 2025) and MMGR (Cai et al., 2025). VRBench comprises five spatial planning tasks: maze navigation (regular, irregular, and 3D variants), trapfield traversal, and Sokoban puzzles. We treat VRBench as an out-of-distribution benchmark, since ReaGen-27k is constructed with layout configurations or difficulty levels distinct from those used in the VRBench test sets. We additionally adopt the indoor navigation task from MMGR. Following their settings, we report Exact Match (EM), Success Rate (SR), and Precision Rate (PR) for VRBench, and Task Completeness (TC) and Instruction Following (IF) for MMGR. The primary metrics are: (a) **EM**. Percentage of videos that perfectly follow the optimal reasoning path. (b) **TC**. Percentage of videos that successfully achieve the task goals. Please refer to Appendix C.1 for task definitions, more metric definitions and benchmark descriptions.

**Baselines**. We compare ReaForest against both closed-source VGMs (*e.g.,* Veo-3.1-fast (DeepMind, 2025), Sora-2 (OpenAI, 2025), Kling-v1 (Kling AI, 2025), and Seedance-1.0-pro (Gao et al., 2025)) and open-source VGMs (Wan2.5-i2v-preview and Wan2.2-TI2V-5B (Wan et al., 2025)). To compare Chain-of-Frame (CoF) with Chain-of-Thought (CoT) reasoning, we also evaluate representative MLLMs including Gemini-2.5-Pro (Comanici et al., 2025), GPT-5-high, and Qwen2.5-VL-7B (Bai et al., 2025) (both base and fine-tuned variants).

**Implementation Details**. We fine-tune Wan2.2-TI2V-5B for five epochs with 100 dataset repeats on our ReaGen-27k. For ReaTTS, we set generation budget $G = 5$ and use Gemini-2.5-Flash as the MLLM verifier with temperature 0.0 for deterministic evaluation. Complete hyperparameter configurations and training details are provided in Appendix C.2.

### 3.2. Main Results and Analyses

The main results are presented in *Table 1* and the key findings are summarized below.

**Overall Performance.** ReaForest achieves substantial improvements over the base model Wan2.2-TI2V-5B and other

*Table 1.* **Left:** Evaluation results on VRBench and MMGR, with the best results in **boldface**. ♠ denotes the primary metric. **Right:** Radar chart of primary-metric performance across six spatial planning tasks (Indoor Navigation from MMGR; others from VRBench).

| Models | VRBench | | | MMGR | |
|---|---|---|---|---|---|
| | EM♠ | SR | PR | TC♠ | IF |
| *Multimodal Large Language Models* | | | | | |
| Gemini-2.5-Pro | 15.9 | 18.1 | 32.6 | - | - |
| GPT-5-high | 17.6 | 27.8 | 10.2 | - | - |
| Qwen2.5-VL-7B | 0.6 | 2.7 | 10.2 | - | - |
| +GRPO | 8.3 | 10.5 | 18.3 | - | - |
| +SFT | 30.5 | 46.0 | 57.8 | - | - |
| *Closed-source Video Generation Models* | | | | | |
| Veo-3.1-fast | 0.6 | 41.4 | 21.7 | - | - |
| Veo-3.1-pro | 1.1 | 46.1 | 27.4 | - | - |
| Sora-2 | 1.4 | 62.2 | 36.8 | 38.1 | 13.6 |
| kling-v1 | 0.0 | 8.9 | 9.2 | - | - |
| Seedance-1.0-pro | 1.1 | 56.1 | 22.7 | - | - |
| MiniMax-Hailuo-2.3 | 0.8 | 56.2 | 22.4 | - | - |
| *Open-source Video Generation Models* | | | | | |
| Wan2.5-i2v-preview | 1.4 | 41.9 | 22.4 | - | - |
| Wan2.2-TI2V-5B | 0.0 | 12.5 | 9.0 | 32.2 | 22.5 |
| Wan-R1 | 37.8 | 82.1 | 68.2 | - | - |
| **ReaForest** | **88.4** | **100.0** | **93.0** | **56.7** | **31.2** |
| Δ | ↑ 88.4 | ↑ 87.5 | ↑ 84.0 | ↑ 24.5 | ↑ 8.7 |

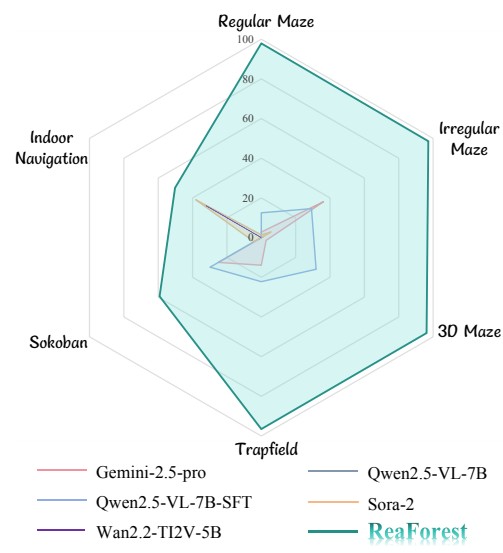

modern video generation models like Sora-2, across all six spatial planning tasks. Notably, ReaForest reaches near-perfect performance on maze navigation variants (regular, irregular, 3D) and trapfield traversal, while also showing significant gains on Sokoban and indoor navigation. This consistent improvement across tasks with diverse reasoning requirements, ranging from continuous pathfinding (maze) to discrete manipulation (Sokoban), validates that the dataset ReaGen-27k activates the foundational reasoning capabilities while the test-time scaling framework ReaTTS effectively amplifies such capabilities.

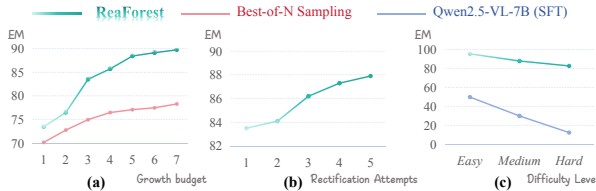

*Figure 3.* Analyses of ReaForest on (a) growth budget, (b) rectification attempts, (c) difficulty level.

**CoF vs CoT.** ReaForest substantially outperforms MLLMs across all tasks, even when the ReaTTS module is removed, validating the superiority of CoF reasoning for vision-centric spatial planning. Under identical training data, fine-tuned Qwen2.5-VL-7B (both SFT and GRPO variants) achieves only moderate improvements, while ReaForest yields significant gains. This difference highlights a

fundamental advantage: video generation naturally tracks evolving visual states across time, making it inherently better suited than text-based reasoning for long-horizon visual reasoning tasks.

**Effectiveness of ReaTTS.** As presented in *Figure 3*, we further validate ReaTTS through two complementary analyses: (1) *Growth Budget.* As shown in *Figure 3a*, performance scales efficiently with generation budget: The primary metric EM of VRBench increases rapidly from low budgets (G=1) and reaches near-optimal performance at moderate budgets (G=5), then plateaus at higher values. In contrast, Best-of-$N$ sampling exhibits slower growth and achieves lower overall performance despite requiring comparable computational budget. This validates that ReaTTS effectively leverages additional test-time computation through adaptive forest growth, achieving superior performance with moderate computational overhead. (2) *Reflective Rectification.* To assess the scalability of reflective rectification, we extend its triggering condition to any hard constraint violation rather than all branches pruned, increase the allowed number of rectification attempts, and fix the growth budget to 3. As can be seen in *Figure 3b*, performance of ReaForest improves progressively with increased rectification attempts, confirming that reflective rectification effectively handles systematic errors through actionable refinement in complex reasoning scenarios. We also provide more comparisons with alternative test-time scaling strategies and other analyses experiments (*e.g.,* efficiency analyses) in Appendix C.3.

*Table 2.* **Left.** Ablation results of ReaForest with the best results in **boldface**. ♠ denotes the primary metric. **Right.** Qualitative results for self-correction, parallel thinking, and scalable reasoning of our ReaForest.

| Method | VRBench | | | MMGR | |
|---|---|---|---|---|---|
| | EM♠ | SR | PR | TC♠ | IF |
| *Reasoning Activation* | | | | | |
| w/o Reasoning Activation | 0.0 | 19.6 | 11.8 | 32.2 | 22.5 |
| w/o Text-Image Alignment | 85.2 | 100.0 | 89.7 | 53.3 | 29.7 |
| *REA-TTS* | | | | | |
| w/o ReaTTS | 70.2 | 95.7 | 78.3 | 45.8 | 28.7 |
| w/o Budget Reallocation | 82.3 | 99.2 | 87.2 | 50.0 | 29.5 |
| w/o Temperature Scaling | 84.1 | 100.0 | 89.4 | 53.3 | 31.2 |
| w/o Temperature Adaptation | 85.3 | 100.0 | 90.7 | 52.5 | 31.2 |
| w/o Forest Entropy | 86.0 | 100.0 | 91.5 | 55.0 | 31.2 |
| w/o Reflective Rectification | 81.5 | 97.9 | 86.9 | 49.2 | 28.1 |
| *Hierarchical Verification* | | | | | |
| w/ Random Verification | 61.6 | 94.5 | 73.8 | 37.5 | 24.1 |
| w/o Hard Constraints | 83.7 | 97.2 | 89.1 | 50.8 | 27.8 |
| w/o Soft Constraints | 85.1 | 100.0 | 90.5 | 55.8 | 31.2 |
| w/o Terminal Constraints | 85.7 | 98.3 | 91.3 | 54.2 | 31.2 |
| **ReaForest** | **88.4** | **100.0** | **93.0** | **56.7** | **31.2** |

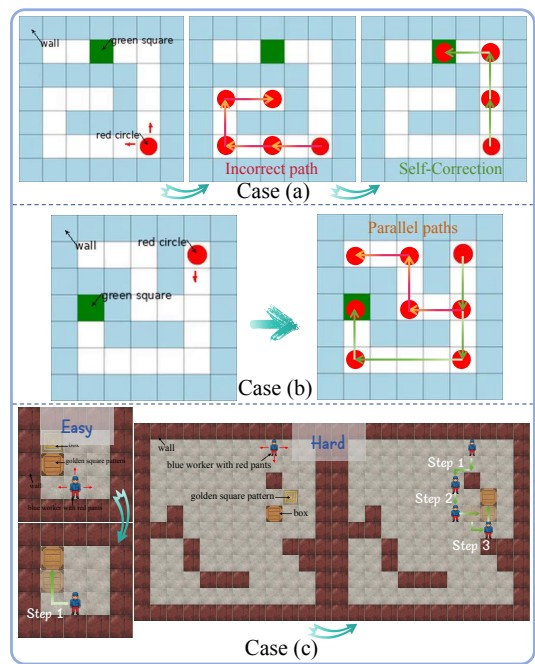

Case (a)

Case (b)

Case (c)

**ReaForest excels at Higher-Difficulty Tasks.** A key strength of ReaForest is its robust performance across increasing task difficulty (*Figure 3c*). While the SFT variant of Qwen2.5-VL-7B degrades significantly with task complexity, dropping from 49.6% EM on easy level to merely 12.2% EM on hard level, ReaForest maintains comparable strong performance across all difficulty levels. This demonstrates the scalability of the CoF reasoning paradigm and further validates ReaForest's effectiveness for vision-centric long-horizon reasoning tasks, particularly in handling challenging scenarios that CoT fails.

## 3.3. Ablation Study

To investigate the contribution of each component, we conduct comprehensive ablations of ReaForest as follows:
**Ablation on Reasoning Activation.** (1) *w/o Reasoning Activation*, which skips fine-tuning the base model on ReaGen-27k; and (2) *w/o Text-Image Alignment*, which uses the raw data without text-image alignment for activation.
**Ablation on ReaTTS.** (1) *w/o ReaTTS*, which eliminates the ReaTTS module; (2) *w/o Budget Reallocation*, which removes the entropy-aware budget reallocation module and fixes all branch budgets to 1; (3) *w/o Temperature Scaling*, which omits the temperature scaling from softmax and directly adopt softmax to compute probabilities distribution; (4) *w/o Temperature adaptation*, which disables temperature adaptation and uses fixed temperature; (5) *w/o Forest Entropy*, which removes forest entropy modulation in the temperature adaptation process; (6) *w/o Reflective Rectification*,

which eliminates the reflective rectification mechanism.
**Ablation on Hierarchical Verification.** (1) *w/ Random Verification*, which replaces the hierarchical constraint verification results with random rating, constraint violation, and terminal satisfaction feedback; (2) *w/o Hard Constraints*, (3) *w/o Soft Constraints*, (4) *w/o Terminal Constraints*, which remove the hard constraints, soft constraints, and terminal constraints, respectively.

The ablation results in *Table 2* reveal several key findings: (a) **Reasoning Activation.** *w/o Reasoning Activation* results in significant performance degradation, confirming that the activation process establishes essential foundational capabilities for generative video reasoning. Furthermore, ReaForest outperforms *w/o Text-Image Alignment*, validating the effectiveness of text-image alignment in incorporating reasoning signals into visual pixel space. (b) **ReaTTS.** ReaForest exceeds *w/o ReaTTS*, validating the utility of ReaTTS in amplifying reasoning capacity at inference time. Moreover, *w/o Budget Reallocation* underperforms ReaForest, indicating that adaptive budget allocation is crucial for efficiently directing computational resources toward promising reasoning paths. ReaForest consistently outperforms both *w/o Temperature Scaling* and *w/o Temperature Adaptation*, demonstrating the effectiveness of adaptive temperature control for balancing exploration and exploitation. The degradation of *w/o Forest Entropy* validates that entropy modulation in temperature adaptation help regulate diversity. Additionally, ReaForest exceeds *w/o Reflective Rectification*, proving the importance of failure recovery in challenging scenarios. (c)

**Hierarchical Verification.** *w/o Random Verification* leads to substantial performance degradation, which indicates that ReaTTS depends on a reliable verification mechanism. Random feedback introduces harmful noise that misdirects budget allocation and branch selection. ReaForest surpasses *w/o Hard Constraints*, *w/o Soft Constraints*, and *w/o Terminal Constraints*, demonstrating that all three constraint types are necessary for providing comprehensive verification and actionable feedback for ReaTTS.

### 3.4. Qualitative Analyses

We observe several emergent behaviors characteristic of CoF reasoning, as illustrated in the right panel of *Table 2*: **Case (a): Self-Correction.** The "red circle" initially commits to an incorrect path but subsequently pivots to the correct trajectory toward the target ("green square") within a single reasoning step. This exhibits self-correction capabilities of ReaForest analogous to the reflection moments observed in CoT reasoning, which can be further amplified through our reflective rectification mechanism. **Case (b): Parallel Thinking.** At a decision point (fork in the path), the "red circle" spawns two distinct candidate trajectories simultaneously. This illustrates ReaForest's inherent capacity for parallel reasoning, enabling it to explore multiple trajectories concurrently. ReaTTS systematically exploits this capability by explicitly spawning multiple candidate branches at each step, enabling comprehensive exploration of the solution space. **Case (c): Scalable Reasoning.** ReaForest successfully solves increasingly complex Sokoban puzzles by allocating additional reasoning steps, validating the scalability of our framework with respect to task difficulty. These behaviors naturally align with our ReaTTS framework design, which effectively amplifies these emergent capabilities during test-time inference, establishing VGMs as flexible visual reasoners. Additional qualitative results in *Figure 1* and Appendix E further demonstrate ReaForest's superiority.

## 4. Conclusion and Future Work

In this work, we present ReaForest, a systematic framework for fostering generative video reasoning in video generation models (VGMs) for spatial planning via (1) constructing a dataset ReaGen-27k with text-image alignment mechanism to activate foundational reasoning capabilities; (2) designing a test-time scaling framework ReaTTS to amplify reasoning capacity at inference time, accompanied by (3) hierarchical constraint verification to provide comprehensive verification and actionable feedback. Extensive experiments on VRBench and MMGR demonstrate that ReaForest achieves significant improvement across six reasoning tasks over the base model Wan2.2-TI2V-5B. These results validate the effectiveness of our ReaForest in activating and amplifying generative video reasoning for spatial planning, marking a

meaningful step toward positioning VGMs as visual mental simulators like humans. Looking ahead, we plan to extend ReaForest to broader real-world domains, including robotic navigation and autonomous driving. By grounding generative video reasoning in increasingly complex and dynamic environments, we aim to further advance VGMs toward serving as practical and general-purpose world simulators, a key research frontier for the community.

## 5. Related Works

### 5.1. Generative Video Reasoning

Video generation models (VGMs) (Zheng et al., 2024; Brooks et al., 2024; Wan et al., 2025; Tong et al., 2026) have demonstrated remarkable progress in diverse applications, including customized video content creation (Zhao et al., 2024) and world modeling (Ha & Schmidhuber, 2018). Recently, Wiedemer et al. introduced Chain-of-Frame (CoF) reasoning, a novel paradigm where VGMs solve visual reasoning tasks (Xu et al., 2025b; He et al., 2025) by generating intermediate reasoning frames analogous to Chain-of-Thought (CoT) in LLMs (Guo et al., 2025a) and MLLMs (Feng et al., 2025; Ouyang et al., 2025a;b; Zeng et al., 2026b). Unlike CoT, which is inherently text-centric, CoF reasoning operates directly in the visual domain, enabling explicit tracking of evolving visual states over long horizons. However, recent works (Cai et al., 2025; Yang et al., 2025; Zhou et al., 2025; Luo et al., 2025; Guo et al., 2025b) current VGMs struggle significantly with CoF reasoning across diverse spatial planning tasks that require long-horizon multi-step reasoning (Yang et al., 2025; Xiao et al., 2025) (*e.g.,* maze navigation, Sokoban puzzles, and trapfield traversal). "How to endow VGMs with robust CoF reasoning abilities for spatial planning" remains an open problem, which our work aims to address.

### 5.2. Test Time Scaling for Video Generation

Test-time scaling, allocating additional inference computation to improve test-time performance, has achieved remarkable success in LLMs (Lightman et al.; Wei et al., 2022; Zhang et al., 2025) and MLLMs (Thawakar et al., 2025; Xu et al., 2025a). However, directly transferring existing test-time scaling methods from LLMs and MLLMs to VGMs encounters several fundamental limitations. For example, Best-of-$N$ (Nakano et al., 2021; Stiennon et al., 2020) Sampling generates $N$ complete videos independently and selects the best via an evaluator or metrics, while simple, it wastes substantial computation on trajectories with early errors that propagate deterministically throughout generation (Liu et al., 2025a). Tree-Based Search methods (Xie et al., 2024; Zhou et al., 2024; Liu et al., 2025a) decompose generation into shorter segments and explore branching trajectories. However, they rely on rigid pruning heuristics

(*e.g.,* top-$k$, hard thresholds), which often lead to premature diversity collapse (Xie et al., 2024; Hao et al., 2024). Moreover, they lack recovery mechanisms that revise intermediate errors when all branches fail systematically-a common occurrence in complex reasoning scenarios. Motivated by these limitations, we propose a novel test-time scaling framework that balances exploration and exploitation while incorporating failure recovery mechanisms.

# Acknowledgements

We thank all anonymous reviewers for their insightful comments. This research was partially supported by the National Natural Science Foundation of China under Grant No. 92470205. Xu Sun is the corresponding author of this paper.

# Impact Statement

Current video generation models exhibit limited Chain-of-Frame (CoF) reasoning capabilities, even struggle with desktop spatial planning tasks such maze navigation. In this work, we identify two key factors underlying this limitation and demonstrate that CoF reasoning provides an effective paradigm for addressing spatial planning tasks, focusing on maze navigation, trapfield traversal, Sokoban puzzles, and other variants. By highlighting the advantages of CoF over text-centric Chain-of-Thought (CoT) in modeling evolving visual states, our findings suggest several broader impacts: (i) accelerating the development of video generation models as effective world simulators, and (ii) motivating hybrid reasoning frameworks that flexibly combine CoF and CoT to better tackle complex visual reasoning problems.

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

# Appendix Overview

# A. More Details about ReaGen-27k

## A.1. Potential Impact and Applications of Text-Image Alignment

Beyond its role in ReaGen-27k, text-image alignment suggests two potential directions for generative video reasoning.

**Enabling Image-to-Video Reasoning.** Text-image aligned visual prompts may elicit reasoning capabilities without textual descriptions. Preliminary observations suggest that annotated images (with object references and reasoning hints) could enable spatial planning even in the absence of explicit task descriptions. This could naturally extend Chain-of-Frame reasoning to image-to-video (I2V) generation models, broadening applicability beyond text-image-to-video architectures. By encoding task specifications directly into visual annotations, future work could explore leveraging I2V models for reasoning tasks, potentially expanding the range of architectures amenable to test-time scaling.

**Interactive Visual Reasoning Interface.** Traditional Chain-of-Thought and Chain-of-Frame reasoning rely primarily on textual prompts for human-AI interaction, limiting users to linguistic expressions of reasoning constraints. Text-image alignment introduces a complementary modality where users could specify task requirements through visual editing tools (*e.g.,* sketching arrows, annotating objects, marking regions). This visual interaction paradigm could offer three key advantages: *(1) Intuitive specification*-spatial constraints would be naturally expressed through visual annotations rather than verbose text; *(2) Precise control*-users could specify exact locations, directions, and object relationships that may be ambiguous in text; *(3) Accessibility*-visual editing could lower barriers for users unfamiliar with prompt engineering (Zeng et al., 2026a; 2025). This fusion of visual and textual modalities could establish a more flexible interface for interactive reasoning, particularly valuable for spatial planning tasks where visual specifications may be more natural than linguistic descriptions.

Together, these extensions position text-image alignment as a foundational technique for advancing multimodal reasoning in video generation, enabling both architectural flexibility (I2V compatibility) and enhanced human-AI interaction through visual specification of reasoning tasks.

## A.2. Dataset Construction, Statistics & Training details

**Dataset Statistics.** Our ReaGen-27k consists of five subtasks: maze navigation (regular, irregular, and 3D mazes), Sokoban puzzles, and trapfield traversal, totaling $27,360$ instances. All instances use configurations distinct from the evaluated VRBench, effectively serving as an out-of-distribution benchmark. Detailed statistics are reported in *Table 3*, and example prompt templates are illustrated in *Figure 13*.

**Text-Image Alignment.** Each instance includes: (a) *object references*, consisting of descriptive text labels paired with arrows indicating the corresponding objects (e.g., "red circle" $\rightarrow$ red circle object); and (b) *reasoning hints*, represented by directional arrows overlaid on the image to indicate feasible motion directions.

**Supervised Fine-Tuning.** Videos are segmented based on difficulty: single clips for easy and medium instances, and three clips for hard instances. Using this processed dataset, we fine-tune Wan2.2-TI2V-5B for five epochs with 100 dataset repetitions, applying LoRA (Hu et al., 2022) (rank 32) to the key attention and feedforward modules (q, k, v, o, ffn.0, ffn.2) of the DiT backbone.

# B. More Details about Hierarchical Constraint Verification

We exemplify some protocols in *Figure 14*, while additional prompts will be provided in the codebase to maintain the readability of the Appendix. These protocols can be customized based on different reasoning tasks and modified manually or by LLMs. We obtain the feedback through MLLM-based question-answering using the defined constraints.

# C. More Experiment Details

## C.1. More Benchmark Details

**Benchmark Descriptions**.

- **VRBench**, which spans five distinct maze type: Regular Maze, Irregular Maze, 3D Maze, Sokoban, and Trapfield, covering a wide spectrum of spatial structures and decision patterns.

*Table 3.* Detailed statistics of ReaGen-27k dataset.

| Task Category | Difficulty Levels | Training Samples | Skin Types |
|---|---|---|---|
| **Regular Maze** | $7 \times 7$ (Grid size) | $120 \times 20$ | 20 |
| | $11 \times 11$ (Grid size) | $120 \times 20$ | 20 |
| | $15 \times 15$ (Grid size) | $120 \times 20$ | 20 |
| **Irregular Maze** | $512 \times 512$ (Pixel size) | $120 \times 12$ | 12 |
| | $768 \times 768$ (Pixel size) | $120 \times 12$ | 12 |
| | $1024 \times 1024$ (Pixel size) | $120 \times 12$ | 12 |
| **3D Maze** | $6 \times 6 \times 5$(Width $\times$ Depth $\times$ Height) | $120 \times 12$ | 12 |
| | $8 \times 8 \times 7$(Width $\times$ Depth $\times$ Height) | $120 \times 12$ | 12 |
| | $10 \times 10 \times 9$(Width $\times$ Depth $\times$ Height) | $120 \times 12$ | 12 |
| **Trapfield** | $5 \times 5$ (Grid size), 20% (Trap ratio) | $120 \times 12$ | 12 |
| | $7 \times 7$ (Grid size), 30% (Trap ratio) | $120 \times 12$ | 12 |
| | $11 \times 11$ (Grid size), 35% (Trap ratio) | $120 \times 12$ | 12 |
| **Sokoban** | $5 \times 5$ (Grid size), 1 (Box number) | $120 \times 20$ | 20 |
| | $8 \times 8$ (Grid size), 1 (Box number) | $120 \times 20$ | 20 |
| | $12 \times 12$ (Grid size), 1 (Box number) | $120 \times 20$ | 20 |

- **MMGR**, which is a comprehensive benchmark suite designed to assess generative reasoning across three complementary domains: Abstract Reasoning (*e.g.,* ARC-AGI, Sudoku), Embodied Navigation (*e.g.,* real-world 3D navigation and localization), and Physical Commonsense (*e.g.,* sports and compositional physical interactions). We adopt the indoor navigation sub-task in our evaluation.

**Task Definitions**.

- **Regular Maze.** The grid-based mazes to evaluate the model's ability to perceive basic maze structures and perform pathfinding under hard wall constraints. This task serves as a foundational testbed for assessing core spatial reasoning and problem-solving capabilities in structured environments.

- **Irregular Maze.** We adopt curve-based and non-uniform path designs instead of regular block-shaped layouts. This design prevents reliance on coordinate-aligned position encoding and rigorously evaluates pure visual perception of spatial layouts, explicitly decoupling visual reasoning from text-based or symbolic reasoning.

- **3D Maze.** By extending mazes into three-dimensional spaces with multi-level structures, this task evaluates stereoscopic spatial perception and cross-dimensional path reasoning, requiring the model to jointly reason over horizontal and vertical navigation.

- **Trapfield Traversal.** Trapfield transforms the walls of traditional mazes into grid-shaped trap regions, shifting the objective from finding feasible paths to avoiding penalized areas. The increased freedom of movement introduces softer constraints, challenging the model's ability to perform risk-aware planning and identify cost-optimal trajectories.

- **Sokoban Puzzles.** The agent must push boxes to target locations under irreversible constraints. This task requires the model to internalize and apply task-specific rules on top of pathfinding, emphasizing long-horizon planning and logical reasoning beyond geometric navigation.

- **Indoor Navigation (Top-down View Real-World Navigation).**

  *Task*: Adopting a fixed bird's-eye perspective, this task focuses on global spatial planning and long-horizon prediction. Models generate navigation trajectories on 2D overhead maps, emphasizing reasoning over global geometry and multi-step pathfinding rather than local, egocentric perception.

**Evaluation Metrics**.

*Table 4.* Comparison of test-time scaling strategies for video generation. ReaTTS uniquely combines adaptive exploration-exploitation balancing with failure recovery mechanisms.

| Method | Generation Type | Branch Selection | Diversity Control | Failure Recovery | Adaptive Allocation | Performance (EM) |
|---|---|---|---|---|---|---|
| Best-of-$N$ | Single-pass | Global (final) | Fixed $N$ | ✗ | ✗ | 77.1 |
| Greedy Search | Multi-step | Top-1 (hard) | None | ✗ | ✗ | 81.7 |
| Beam Search | Multi-step | Top-$k$ (hard) | Fixed $k$ | ✗ | ✗ | 83.6 |
| **ReaTTS (Ours)** | Multi-step | Softmax (soft) | Dynamic | ✓ | ✓ | 88.4 |

- **Exact Match (EM).** This metric measures whether the model successfully generates the complete and correct trajectory that aligns with the shortest optimal valid path. One step of deviation from the optimal solution is considered incorrect.

- **Success Rate (SR).** SR measures whether the generated trajectory successfully reaches the designated goal region. It reflects the model's capability to complete the task by arriving at the target position, with a value of 1 indicating successful goal attainment and 0 indicating failure to reach the goal.

- **Precision Rate (PR).** PR quantifies the proportion of consecutively correct steps along the optimal path. It offers a softer metric than EM, reflecting the model's ability to make steady, meaningful progress toward the complete correct trajectory.

- **Task Completeness (TC),** which measures navigation success through two components: (1) *Success Score*—binary indicator (1/0) of whether the agent's final position lies within the designated goal region in the 2D overhead map; (2) *Oracle Success Score*—provides partial credit (1/0) if the agent's trajectory ever intersects or touches the goal region during navigation, even without stopping there. TC evaluates both precise goal achievement and approximate goal-directed progress.

- **Instruction Following (IF),** which evaluates comprehensive adherence to destination integrity (the target region must remain unchanged and be reached without hallucinated alternatives) and scene consistency (the environment must remain static without objects appearing, disappearing, or deforming). This metric ensures the model maintains both goal fidelity and physical plausibility throughout navigation.

## C.2. More Implementation Details

**Training details.** We applied LoRA (Hu et al., 2022) (rank 32) to key attention and feedforward modules (q, k, v, o, ffn.0, ffn.2) of the DiT backbone. The model was trained for 5 epochs, with a dataset repetition factor of 100. We fine-tune Qwen2.5-VL-7B with supervised fine-tuning (SFT) for 10 epochs, using a batch size of 64 and a maximum sequence length of $1,024$. For GRPO training, we employ a rollout group size of 8, a batch size of 8, and a maximum sequence length of $4,000$.

**ReaTTS Configuration.** We utilize Kimi K2 to rectify the conditions for reflective rectification. The rectification prompt is exemplified in *Figure 12*. For easy and medium difficulty levels, we fix reasoning step at 1, and 3 for hard difficulty level. The initial temperature $\tau_0$ is set to $\sqrt{N}$, N is the number of soft constraints.

**Hardware Usage.** We conduct training and evaluation on 16 A800 80GiB GPUs and 8 H20 141GiB GPUs.

*Table 5.* Performance comparison with different verifier and rectifier choices on VRBench.

| Verifier | Deployment | Rectifier | Deployment | EM (%) |
|---|---|---|---|---|
| Gemini-2.5-Flash | API | Kimi-k2 | API | **88.4** |
| Qwen3-VL-235B-A22B | API | Kimi-k2 | API | 87.6 |
| Qwen3-VL-32B | Local | Kimi-k2 | API | 86.3 |
| Qwen3-VL-8B | Local | Kimi-k2 | API | 84.1 |
| Qwen3-VL-8B | Local | Qwen3-32B | Local | 83.7 |

### C.3. More Experiments

**Comparison with Alternative Test-Time Scaling Methods.** *Table 4* presents a comprehensive feature and performance comparison between ReaTTS and existing test-time scaling strategies. For fair comparison, all methods use a total generation budget of 5 clips/videos, with Beam Search employing top-3 selection for branch pruning. Results demonstrate that ReaTTS consistently outperforms all baselines, validating the effectiveness of adaptive exploration-exploitation balancing and reflective rectification in amplifying reasoning capabilities.

**Robustness to Verifier and Rectifier Choices.** To validate the generalizability of ReaTTS, we evaluate performance with different verifier and rectifier implementations. *Table 5* presents results using various MLLM backbones and deployment configurations. ReaForest maintains strong and comparable performance across all settings (83.7%–88.4% EM), demonstrating two key properties: (1) *Framework scalability*: ReaTTS generalizes across diverse verifier-rectifier combinations without requiring specific implementations; (2) *Verification efficiency*: Our hierarchical constraint verification substantially reduces evaluation complexity through structured decomposition-even smaller locally-deployed models (Qwen3-VL-8B: 84.1%) achieve strong performance, whereas holistic video scoring typically requires frontier models. This validates that explicit constraint satisfaction checking is more efficient and accessible than end-to-end heuristic scoring.

*Table 6.* Efficiency analysis on VRBench. Inference time is measured in seconds per sample on H20 (141GB).

| Method | Time (s/sample) | EM (%) |
|---|---|---|
| Best-of-1 | 92.6 | 70.2 |
| Best-of-3 | 252.8 | 75.0 |
| Best-of-5 | 416.7 | 77.1 |
| ReaForest ($G = 1$) | 95.7 | 73.5 |
| ReaForest ($G = 3$) | 261.9 | 83.5 |
| ReaForest ($G = 5$) | 431.7 | 88.4 |

**Efficiency analysis.** As shown in *Table 6*, ReaForest substantially outperforms Best-of-N sampling at comparable costs. With similar inference time ( 430s), ReaForest ($G = 5$) achieves 88.4% EM versus Best-of-5's 77.1%-a +11.3% improvement. This advantage remains consistent across budgets (+8.5% at $G = 3$, +3.3% at $G = 1$), validating that adaptive exploration with reflective rectification enables more effective reasoning than independent sampling.

## D. Theoretical Analyses of ReaTTS

We provide theoretical justification for the entropy-aware budget allocation mechanism in ReaTTS, focusing on the design rationale behind Equations (2) and (3).

### D.1. Information-Theoretic Justification for Temperature-Scaled Softmax

We model branch selection as a decision problem under uncertainty, where the goal is to allocate limited budget $G$ to maximize expected reasoning quality.

**Proposition D.1** (Maximum Entropy Distribution with Moment Constraints)**.** *Given verifier ratings* $\mathbf{r} = (r_1^s, \ldots, r_{|\mathcal{B}_s|}^s)$, *the distribution* $\mathbf{p}$ *that maximizes entropy* $H(\mathbf{p}) = -\sum_i p_i^s \log p_i^s$ *subject to:*

1. *Normalization:* $\sum_i p_i^s = 1$

2. *Moment constraint:* $\sum_i p_i^s r_i^s = \bar{r}$ *for some target expected rating* $\bar{r}$

*is the Gibbs distribution:*

$$p_i^s = \frac{\exp(r_i^s/\tau_s)}{\sum_k \exp(r_k^s/\tau_s)}. \tag{7}$$

*Proof.* Motivated by the maximum entropy principle (jay, 1957), we seek the distribution with maximum entropy:

$$\max_{\mathbf{p}} H(\mathbf{p}) = -\sum_i p_i^s \log p_i^s \tag{8}$$

subject to:

$$\sum_i p_i^s = 1 \quad \text{(normalization)} \tag{9}$$

$$\sum_i p_i^s r_i^s = \bar{r} \quad \text{(moment constraint)} \tag{10}$$

The Lagrangian is:

$$\mathcal{L} = -\sum_i p_i^s \log p_i^s - \lambda_0 \left( \sum_i p_i^s - 1 \right) - \lambda_1 \left( \sum_i p_i^s r_i^s - \bar{r} \right) \tag{11}$$

Taking derivative with respect to $p_i^s$:

$$\frac{\partial \mathcal{L}}{\partial p_i^s} = -\log p_i^s - 1 - \lambda_0 - \lambda_1 r_i^s = 0 \tag{12}$$

Solving:

$$p_i^s = \exp(-1 - \lambda_0 - \lambda_1 r_i^s) = C \exp(-\lambda_1 r_i^s) \tag{13}$$

Setting $\tau_s = -1/\lambda_1$ (temperature interpretation) and normalizing:

$$p_i^s = \frac{\exp(r_i^s/\tau_s)}{\sum_k \exp(r_k^s/\tau_s)} \tag{14}$$

**Interpretation**: Among all distributions consistent with the rating information (via moment constraint $\bar{r}$), softmax is the least committal (maximum entropy), incorporating ratings while maintaining maximum uncertainty about which specific branch to select. $\qquad\square$

### D.2. Adaptive Temperature Scheduling

The temperature schedule (Eq. (3)) combines two complementary adaptation mechanisms:

**Exponential Annealing: Temporal Progression.** The decay term $\exp(s/S)$ implements a time-dependent cooling schedule analogous to simulated annealing in optimization.

**Lemma D.2** (Temporal Annealing). *Under the assumption that the marginal value of exploration decreases as reasoning progresses (*i.e., *branches differentiate over time), exponential annealing $\tau_s \propto \exp(-s/S)$ ensures:*

- *Early stages ($s \ll S$): High temperature maintains diverse exploration when branch quality is uncertain.*

- *Late stages ($s \to S$): Low temperature enables exploitation of clearly superior branches.*

- *Smooth transition: Continuous decay avoids abrupt strategy changes.*

**Entropy Modulation: State-Dependent Adaptation.** The multiplicative factor $(1 + \tilde{H}_s)$ provides local adaptivity based on current forest state, where normalized entropy:

$$\tilde{H}_{s-1} = -\frac{1}{\log |\mathcal{B}_{s-1}|} \sum_{k=1}^{|\mathcal{B}_{s-1}|} p_k^{s-1} \log p_k^{s-1} \in [0, 1] \tag{15}$$

quantifies uncertainty in branch selection.

**Theorem D.3** (Entropy-Aware Adaptation). *The entropy-modulated temperature $\tau_s = \tau_0 \cdot (1 + \tilde{H}_{s-1})/\exp(s/S)$ achieves state-dependent exploration-exploitation balance:*

1. ***High entropy** ($\tilde{H}_{s-1} \to 1$): Uniform rating distribution indicates competitive branches. Increasing $\tau_s \approx 2\tau_0/\exp(s/S)$ sustains exploration by flattening the softmax distribution.*

2. **Low entropy** ($\tilde{H}_{s-1} \to 0$): *Peaked distribution indicates a clear winner. Reducing $\tau_s \approx \tau_0 / \exp(s/S)$ enables aggressive exploitation by sharpening allocation toward top branches.*

*Proof.* Consider the allocation ratio between the best and second-best branches:

$$\frac{p_{\text{best}}^s}{p_{\text{2nd}}^s} = \exp\left(\frac{r_{\text{best}}^s - r_{\text{2nd}}^s}{\tau_s}\right). \tag{16}$$

When entropy is high (small rating gap $\Delta r = r_{\text{best}}^s - r_{\text{2nd}}^s$), modulation increases $\tau_s$, yielding smaller ratio (more uniform allocation). When entropy is low (large $\Delta r$), smaller $\tau_s$ amplifies the ratio (concentrated allocation). This state-dependent adjustment prevents both premature convergence (high entropy regime) and inefficient exploration (low entropy regime).  □

### D.3. Comparison with Alternative Strategies

We compare our approach with common baselines:

*Table 7.* Comparison of budget allocation strategies.

| Strategy | Exploration | Exploitation | Adaptivity |
|---|---|---|---|
| Fixed top-$k$ | ✗ | ✓ | ✗ |
| Uniform sampling | ✓ | ✗ | ✗ |
| UCB-style | ✓ | ✓ | ✓ (count-based) |
| **ReaTTS (Ours)** | ✓ | ✓ | ✓ (state-based) |

**Key distinctions:**

- **Fixed top-$k$** (Freitag & Al-Onaizan, 2017): Hard thresholding causes discontinuous allocation and eliminates potentially promising branches prematurely.

- **Uniform sampling** (Wang et al., 2022): Ignores rating information, leading to inefficient budget allocation.

- **UCB-style** (Upper Confidence Bound) (Auer et al., 2002): Requires explicit visit counts and assumes stationary rewards, unsuitable for evolving video generation.

- **ReaTTS**: Soft allocation with dual temporal and state-dependent adaptation, enabling smooth exploration-exploitation transition without explicit counting.

### D.4. Summary

Our entropy-aware budget reallocation strategy is grounded in:

1. **Information-theoretic optimality**: Softmax allocation maximizes entropy-regularized utility (Proposition D.1).

2. **Dual adaptation**: Exponential annealing handles temporal progression (Lemma D.2), while entropy modulation provides state-dependent control (Theorem D.3).

3. **Practical advantages**: Avoids discontinuities of hard selection and inefficiencies of count-based methods.

These theoretical insights validate the design choices in ReaTTS and establish a principled foundation for test-time scaling in generative video reasoning.

## E. More Qualitative Analyses

We provide more qualitative cases to further illustrate the superiority of our ReaForest and reveal some remained shortcomings of our framework to motivate future work. We also provide demo videos in Supplementary Materials.

### E.1. Success cases

*Figure 4–Figure 9* present qualitative comparisons between ReaForest and the Wan2.2-TI2V-5B baseline across diverse reasoning tasks. These examples showcase ReaForest's consistent ability to generate valid trajectories:

**Regular Maze** (*Figure 4*): ReaForest successfully navigates through grid-based corridors with precise obstacle avoidance, while the baseline generates invalid paths.

**Irregular Maze** (*Figure 5*): ReaForest demonstrates pure visual spatial reasoning on curved, non-uniform layouts without relying on coordinate-based heuristics. The baseline fails to maintain valid trajectories through these unstructured environments.

**3D Maze** (*Figure 6*): ReaForest exhibits stereoscopic spatial perception, successfully navigating multi-level structures with vertical transitions. In contrast, the baseline produces visual hallucinations (golden spheres) and invalid navigation paths.

**Trapfield Traversal** (*Figure 7* and *Figure 8*): ReaForest consistently generates risk-aware trajectories that minimize trap contact across diverse configurations, demonstrating effective planning under soft penalty constraints. The baseline fails to balance goal progress against hazard avoidance, producing suboptimal paths with visual artifacts.

**Sokoban Puzzle** (*Figure 9*): ReaForest successfully pushes boxes to target locations through valid sequential moves, exhibiting causal foresight and deadlock avoidance. The baseline generates physically impossible box manipulations and visual distortions that violate game rules.

### E.2. Failure cases

We present several failure cases to reveal potential improvement directions for ReaForest.

**Case 1**: Minor Physical Constraint Violations. As shown in *Figure 10*, although ReaForest generates the correct reasoning path and achieves the task objective, the final frame exhibits a minor overlap between the box and the wall. Compared to the ground truth, this indicates that while the overall trajectory is correct, fine-grained physical constraints are not perfectly satisfied, suggesting room for improvement in spatial precision.

**Case 2**: Visual Ambiguity and Model Capacity Limitations We present a failure case with growth budget of 3 in *Figure 11*. As we can see, three reasoning branches that all converge to incorrect paths due to the stacked ladders create visual illusions that challenge accurate spatial perception. Even when reflective rectification identifies constraint violations and modifies the condition prompt, the model continues generating incorrect trajectories. This failure is primarily attributed to the limited capabilities of Wan2.2-TI2V-5B, a relatively weak generation model. Due to the scarcity of open-source text-image-to-video models, we are constrained to use this base model. We are confident that with more powerful open-source models in the future, ReaForest will achieve significantly better performance on such challenging cases.

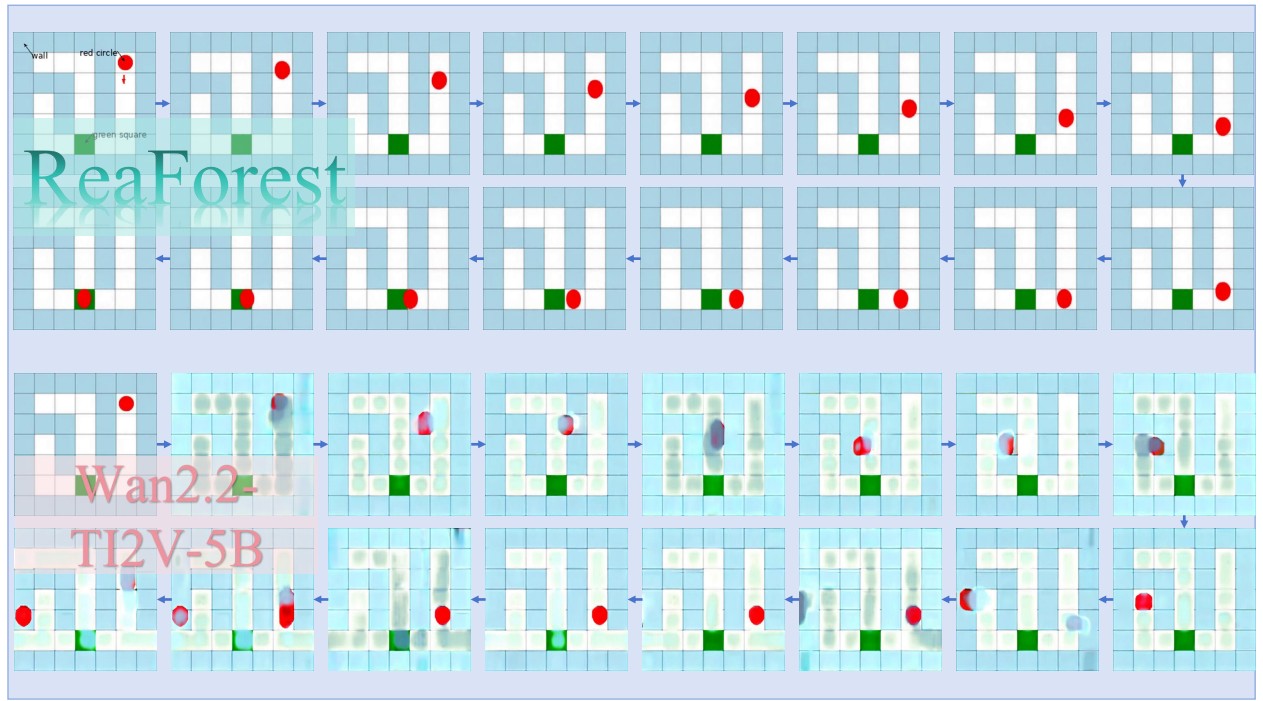

*Figure 4.* Regular Maze navigation. **Top**: ReaForest generates a valid trajectory from start (red circle) to goal (green square), successfully navigating the grid-based corridors while avoiding walls. **Bottom**: Wan2.2-TI2V-5B fails in generating a valid trajectory.

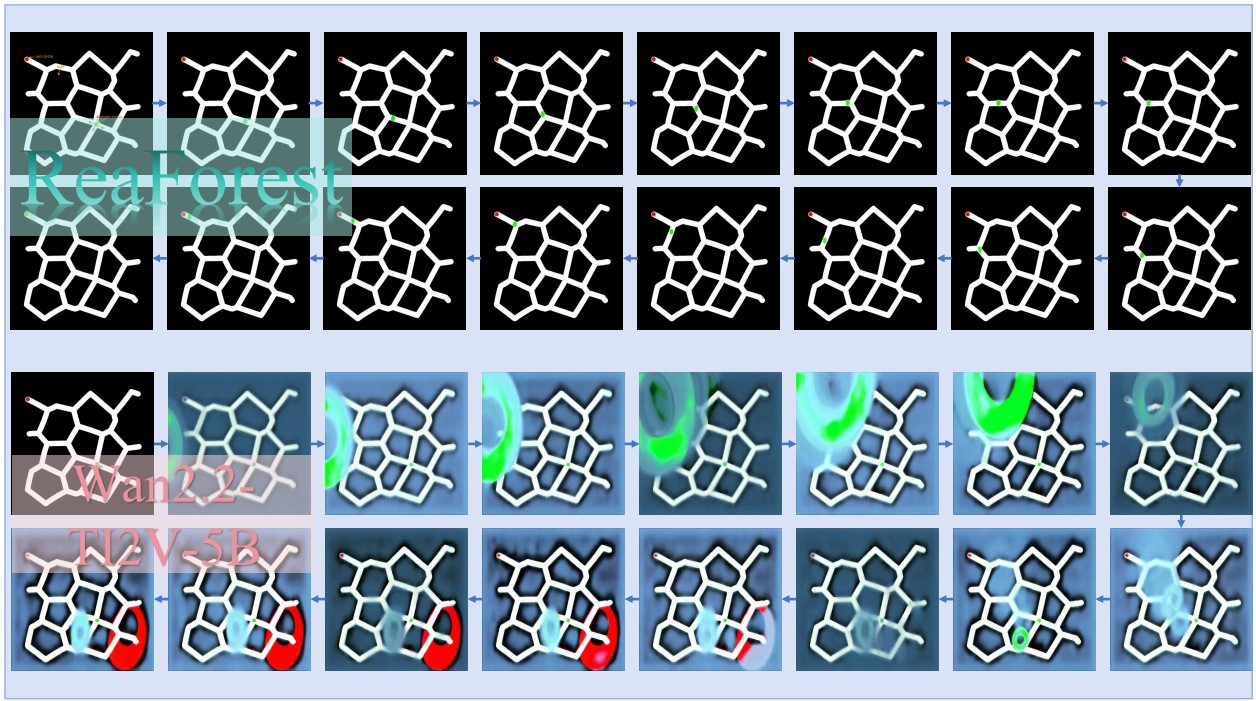

*Figure 5.* Irregular Maze navigation. **Top**: ReaForest generates a valid trajectory through curved corridors. **Bottom**: Wan2.2-TI2V-5B fails, illustrating the advantage of ReaForest in handling non-grid spatial layouts.

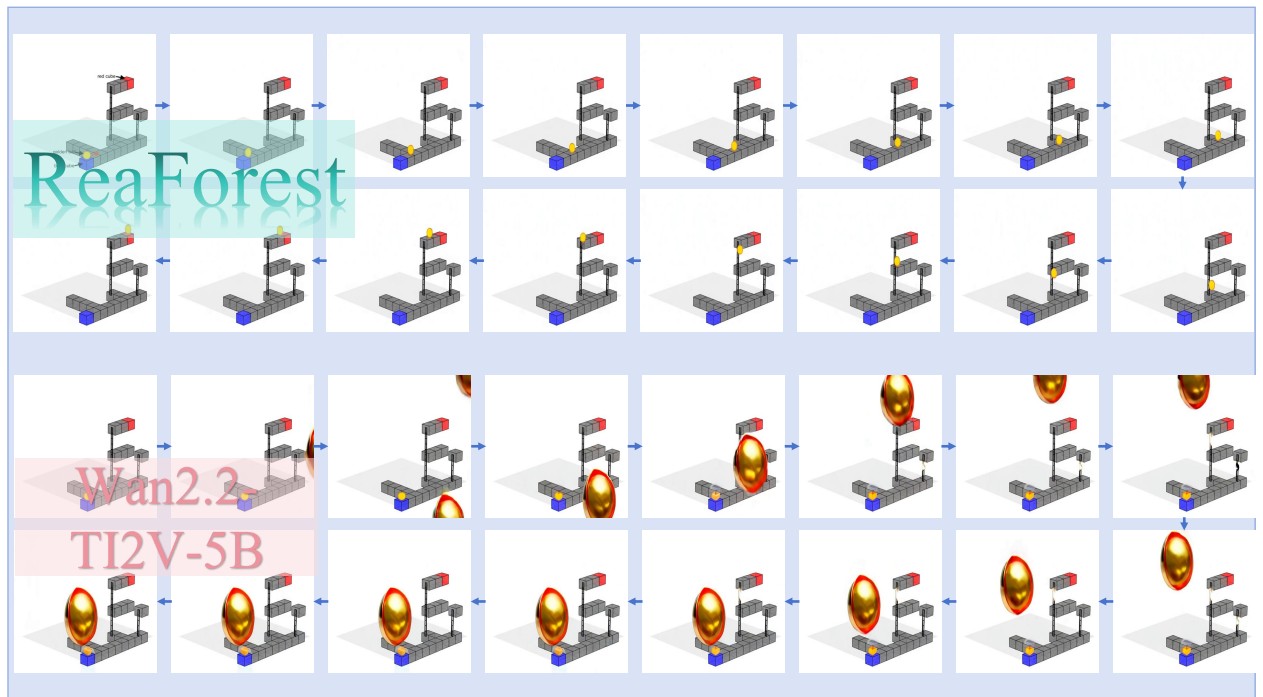

*Figure 6.* 3D Maze navigation. **Top**: ReaForest successfully navigates through multi-level three-dimensional structures. **Bottom**: Wan2.2-TI2V-5B generates invalid trajectories with visual artifacts.

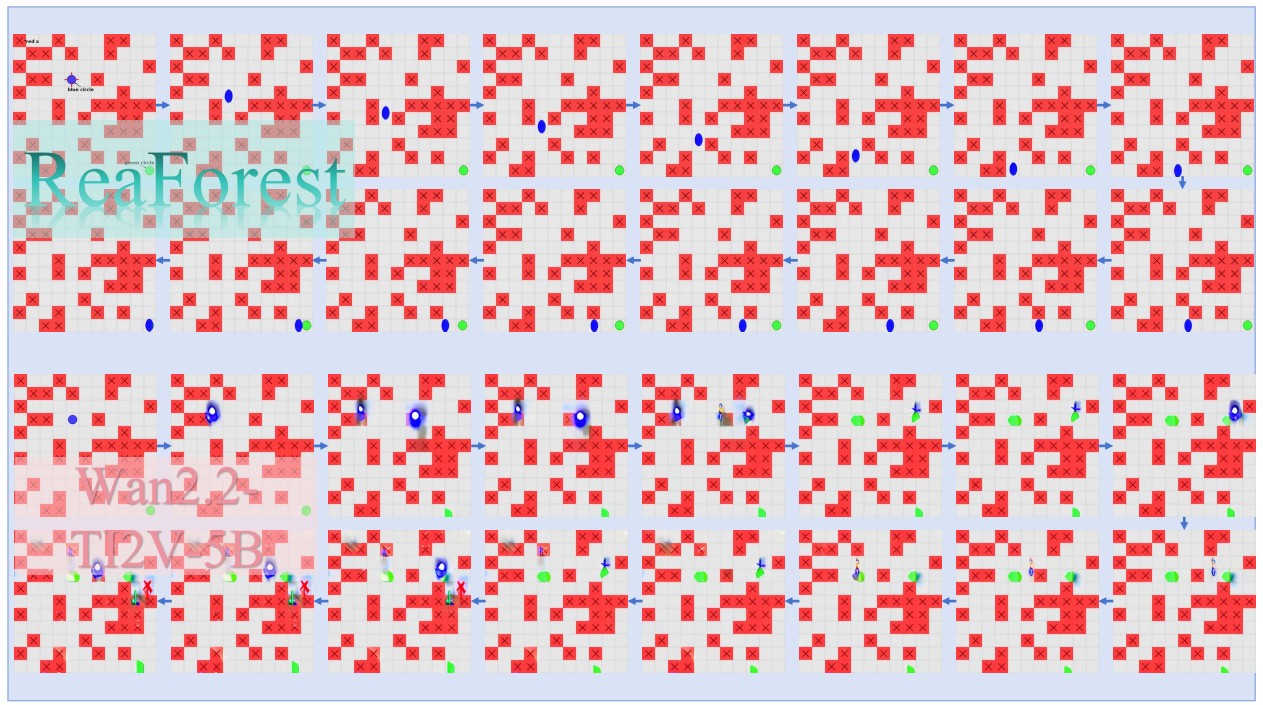

*Figure 7.* Trapfield traversal. **Top**: ReaForest generates a obstacle-aware trajectory that minimizes trap contact. **Bottom**: Wan2.2-TI2V-5B fails to avoid hazards effectively.

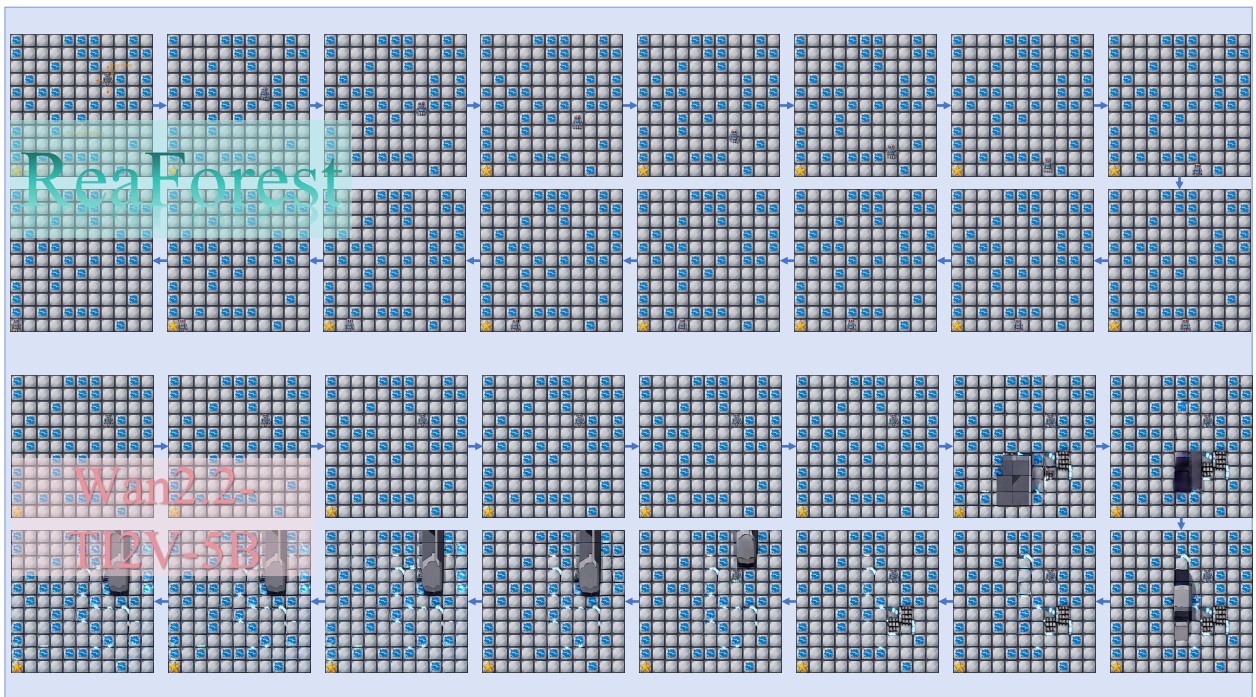

*Figure 8.* Additional qualitative comparison on Trapfield traversal. **Top**: ReaForest consistently generates obstacle-aware trajectories across diverse trap configurations. **Bottom**: Wan2.2-TI2V-5B produces suboptimal paths with visual artifacts.

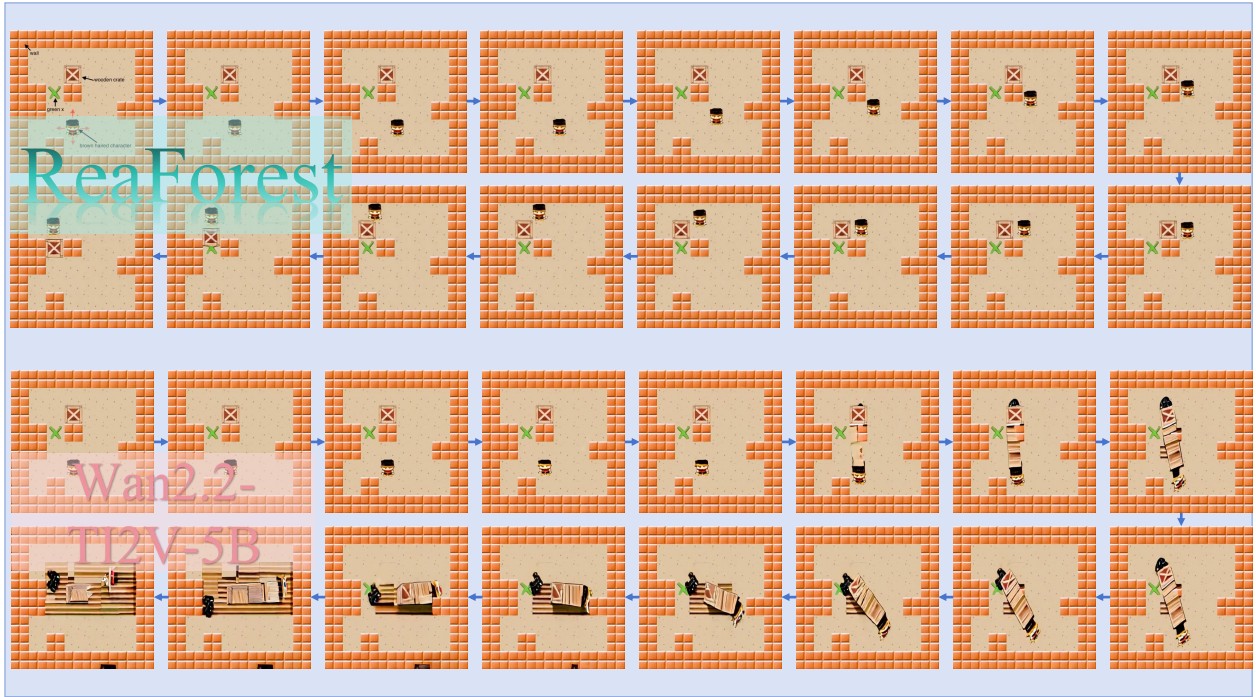

*Figure 9.* Sokoban puzzle. **Top**: ReaForest successfully pushes boxes to target locations with valid sequential moves. **Bottom**: Wan2.2-TI2V-5B generates invalid trajectories with visual distortions and incorrect box movements.

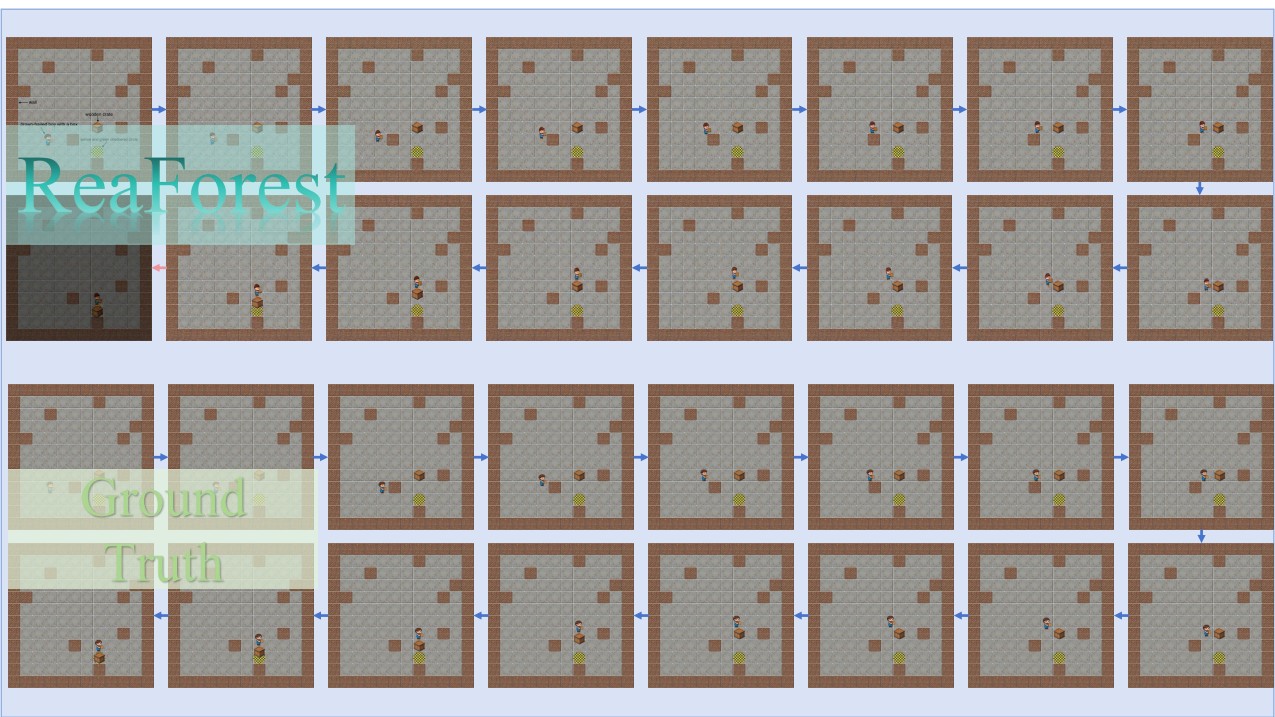

*Figure 10.* Failure Case 1. ReaForest generates the correct reasoning path (top row) and achieves the task objective, but the final frame shows minor box-wall overlap compared to the ground truth (bottom row), indicating imperfect satisfaction of fine-grained physical constraints.

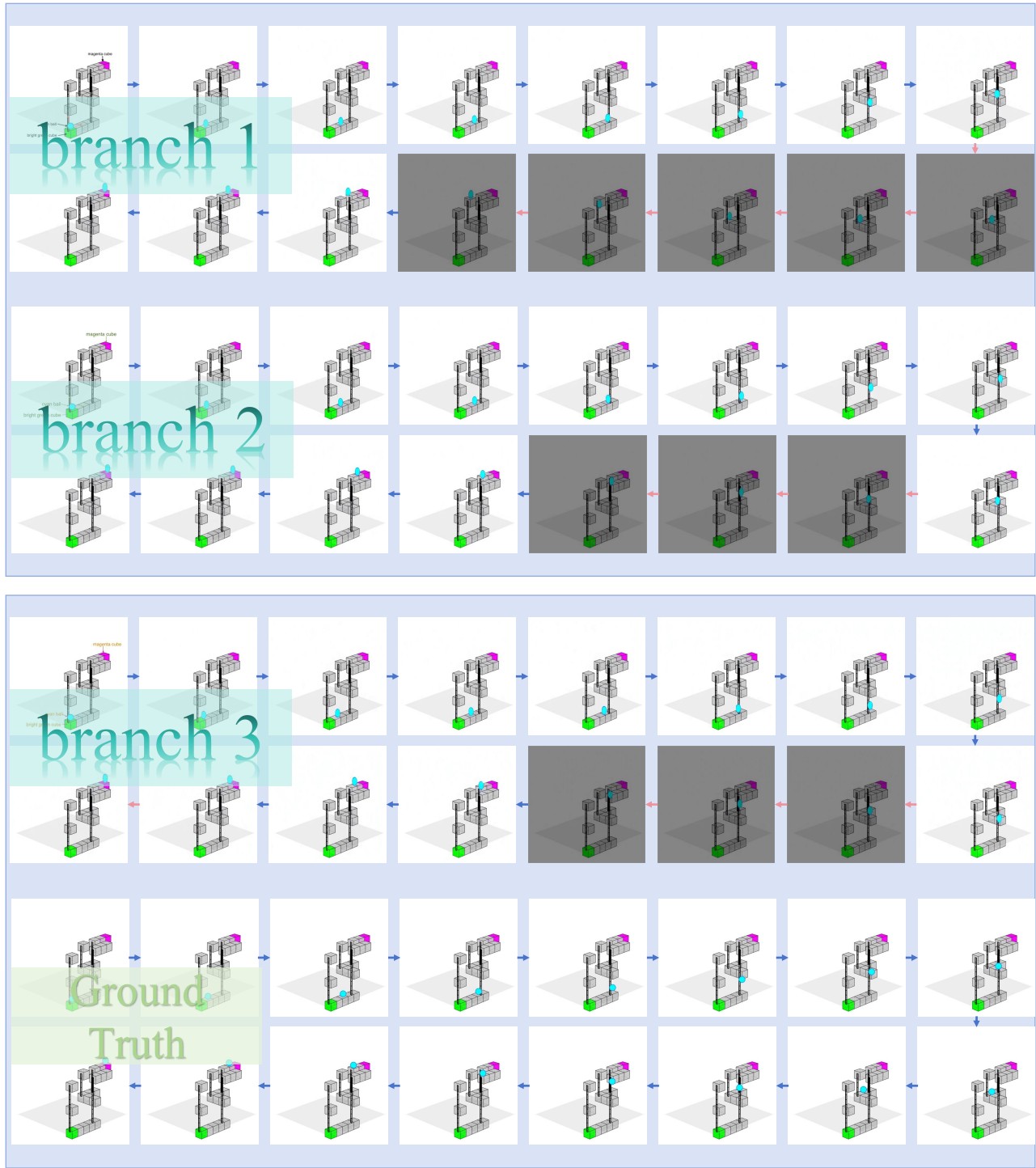

*Figure 11.* Failure Case 2: All three reasoning branches fail due to visual ambiguities from stacked ladders. Gray backgrounds indicate incorrect trajectories. Despite reflective rectification, all branches fail due to the base model's limited capacity to handle visual illusions.

You are an expert at analyzing video generation failures and improving task descriptions to emphasize constraint satisfaction.

Given:
1. Original Task Description: {original_task_description}
2. Violated Constraints: {violated_constraints}

Your task: Generate an augmented task description that:
- Maintains the original task objective
- Explicitly emphasizes avoiding the identified constraint violations
- Adds specific warnings and preventive instructions
- Uses clear, actionable language

Requirements:
- Start with the original task
- Add explicit constraint emphasis
- Provide specific spatial/behavioral guidance
- Keep the description concise but comprehensive

Now generate the augmented task description:
Original Task: {original_task_description}
Violated Constraints: {violated_constraints}

Augmented Task Description:

*Figure 12.* Example rectifier prompt.

| Regular/Irregular Maze | Create a 2D animation based on the provided image of a maze.
The {player} slides smoothly along the {floor} path, stopping perfectly on the {goal}.
The {player} never slides or crosses into the wall of the maze.
The camera is a static, top-down view showing the entire maze. |
|---|---|
| 3D Maze | Create a 3D animation based on the provided image of a cube maze.
A {player} slides smoothly along the {floor} pathway, climbs up the vertical ladders step by step, and finally stops perfectly on the {goal} at the top.
The {player} never touches or passes through the {start} or out of cube areas of the maze.
The camera remains static in an isometric, top-down angle showing the entire structure. |
| Trapfield | Create a 2D animation based on the provided image of a maze.
The {player} slides smoothly along the {floor} path, stopping perfectly on the {goal}.
The {player} never slides into or crosses the {trap}.
The camera is a static, top-down view showing the entire maze. |
| Sokoban | Create a 2D animation based on the provided image of a grid puzzle.
The {player} moves into position behind the {box} and smoothly pushes it toward the {goal}.
The {box} only slides when pushed frombehind by the {player} and moves in a straight line along the {floor}.
When the direction of the {box}'s movement needs to change, the {player} must reposition itself to a new side of the {box}.
The {box} never crosses or overlaps any walls. |

*Figure 13.* Prompt templates for different tasks.

| | |
|---|---|
| Hard Constraints | - "Is the white rabbit / orange carrots visible and continuously present across all frames of the video, without disappearing or being replaced? If yes output 1 else output 0. The output is:"
- "Does the white rabbit crosses the orange carrots in any frame of the video. If yes output 0 else output 1. The output is:"
- "Does either object disappear, flicker, or reappear abruptly in a different location at any point in the video? If no output 1 else output 0. The output is:"
... |
| Soft Constraints | - "Assess how smooth and physically continuous the motion of the white rabbit is across frames. Output an integer score from 1 to 5 (1 = very jittery or discontinuous, 5 = perfectly smooth). The output is:"
- "Assess whether the spatial relationship between the white rabbit and the orange carrots evolves consistently toward the intended interaction. Output an integer score from 1 to 5 (1 = inconsistent or random, 5 = fully coherent). The output is:"
- "Assess the stability of the background and absence of distracting or irrelevant objects. Output an integer score from 1 to 5. The output is:"
... |
| Terminal Constraints | - "In the final frame, is the white rabbit touching or overlapping the orange carrots? If yes output 1 else output 0. The output is:"
- "After reaching the orange carrots, does the white rabbit remain at the target location? If yes output 1 else output 0. The output is:"
... |

*Figure 14.* An example for hierarchical constraint verification protocols of regular maze.

