# OpenReview forum: "ReaForest: Fostering Generative Video Reasoning for Spatial Planning"
_ICML.cc/2026/Conference — ICML 2026 regular_

### Official Review · Reviewer_CFzz · 2026-02-25

**Soundness:** 3
**Presentation:** 3
**Significance:** 3
**Originality:** 2
**Overall Recommendation:** 4
**Confidence:** 3

**Summary:**

This manuscript proposes ReaForest, which aims to address the lack of logical reasoning capabilities in video generative models for complex spatial planning tasks. The authors contribute the ReaGen-27k dataset, designed to activate reasoning through visual signals, and the ReaTTS inference-time scaling framework, which achieves performance gains via branch evolution and rectification. Experimental results demonstrate that ReaForest achieves significant improvements over the base model, Wan2.2-TI2V-5B, on benchmarks such as VRBench and MMGR.

**Compliance With Llm Reviewing Policy:**

Affirmed.

**Final Justification:**

I appreciate the authors' rebuttal clarification, i will keep my original positive score.

**Key Questions For Authors:**

- Can the author provide the success rate of Reflective Rectification in successfully recovering correct paths after failure?
- Is the text-image alignment (visual guidance) mechanism significantly modified compared to In-Video Instructions [2]? If so, what are the specific changes? Additionally, does this method introduce potential artifacts or visual interference in the generated videos?
- How does the system's performance degradation curve look if a smaller, locally-deployed verifier (e.g., Qwen3-VL) is used instead of frontier models?

*For detailed questions, please refer to the Weaknesses.*

---
[2] In-Video Instructions: Visual Signals as Generative Control

**Limitations:**

While the authors discuss limitations regarding model capability, they do not adequately address the drawbacks of high time costs and computational resource requirements in real-time interactive applications.

**Strengths And Weaknesses:**

**Stengths**
- Effective mechanism: the introduction of information-theoretic Entropy-Aware Budget Reallocation provides a more principled mathematical framework for branch searching in video generation compared to traditional Top-K pruning, offering a systematic way to balance exploration and exploitation.
- ReaGen-27k: this work constructs a video reasoning dataset, ReaGen-27k, covering a wide range of scenarios requiring multi-step logical planning.

**Weaknesses**
- **Restricted domains:** evaluation is limited to structured environments; generalization to natural, non-rule-based physics remains unproven.
- **Significant overhead:** efficiency analysis in Tab6 shows that the inference time for ReaForest is 4.6 times that of single-pass generation (431.7s vs. 92.6s). This magnitude of overhead poses a significant challenge to the feasibility of practical applications.
- **Verifier dependency:** the success of the search loop depends heavily on the accuracy of the external MLLM verifier. There is a lack of quantitative analysis on how hallucinations from the verifier might interfere with the entire forest evolution process.
- **Missing discussion:** several related benchmarks and TTS strategies have recently emerged in the video generative reasoning, such as VideoTPO strategy in TiViBench[1]. ReaForest fails to discuss the trade-offs between its reasoning path exploration route and these iterative prompt optimization TTS approaches in terms of performance and efficiency, limiting the understanding of its unique contributions.
- **Rigor of OOD evaluation:** while the layouts in VRBench differ from the training set, the task paradigm (e.g., mazes) remains highly similar. It is possible that the model has learned a specific rule-based visual search algorithm rather than a generalized capacity for logical planning.

---
[1] TiViBench: Benchmarking Think-in-Video Reasoning for Video Generative Models, CVPR 2026

---

> ### Author Rebuttal · Authors · 2026-03-30
>
> We sincerely appreciate your insightful comments and addressed each concern below.
>
> > **W1 & W5:** Restricted domains and OOD evaluation.
>
> **R1:** We intentionally adopt structured environments as a controlled testbed for studying generative video reasoning, as current VGMs still struggle with fundamental spatial planning tasks (e.g., maze navigation). This setting enables precise and controlled evaluation of reasoning capabilities under well-defined constraints.
> In addition, ReaForest achieves a +24.5% TC improvement on MMGR, a more realistic benchmark with higher visual complexity and less structured layouts, indicating that the observed gains extend beyond purely synthetic settings to more challenging OOD scenarios.
>
> We acknowledge that generalization to natural physical environments remains an open question. We view this as an important direction for future work, including extending VGMs to more complex real-world settings such as robotic navigation in physical environments.
>
> > **W2 & L1:** Significant overhead.
>
> **R2:** ReaForest offers a flexible cost–performance tradeoff through the generation budget G. As shown in Table 6 of the Appendix, ReaForest (G=3) achieves 83.5% EM with a runtime of 261.9s per sample, outperforming Best-of-5 (77.1% EM, 416.7s) while reducing inference cost by 37%. This demonstrates that users can adjust the growth budget according to latency constraints without substantial degradation in reasoning performance.
>
> > **W3:** Lack of quantitative analysis on how hallucinations from the verifier might interfere with the entire forest evolution process.
>
> **R3:** We thank the reviewer for this concern. To evaluate robustness to verifier hallucinations, we test ReaTTS with MLLM verifiers of varying capacities, as reported in Table 5 of the Appendix. The results show that ReaTTS consistently maintains strong performance across different verifiers, with only moderate variation, indicating robustness to potential verifier hallucinations.
>
> > **W4:** Missing discussion with VideoTPO strategy.
>
> **R4:** Thanks for your advice, we will add comparison with VideoTPO to Section 5.2 and extend Table 4 to include VideoTPO as a baseline.
>
> Conceptually, VideoTPO and ReaTTS operate in different spaces: VideoTPO performs iterative optimization in the prompt space based on holistic video-level feedback, whereas ReaTTS conducts trajectory-level test-time search with step-wise constraint verification and targeted failure correction. This makes ReaTTS particularly suitable for multi-step spatial planning, where early-stage errors must be detected and corrected to prevent compounding failures.
> Moreover, VideoTPO is training-free and therefore bounded by the base model’s inherent capability, while ReaForest combines training-time reasoning activation with test-time scaling, leading to a fundamental expansion of spatial reasoning ability, as evidenced by the base model achieving 0.0% EM without fine-tuning regardless of inference strategy.
>
> > **Q1:** Can the author provide the success rate of Reflective Rectification in successfully recovering correct paths after failure?
>
> **R5:** Reflective Rectification achieves a 76.5% success rate in recovering correct trajectories after failure, demonstrating its effectiveness as an error-correction component in ReaTTS.
>
> > **Q2:** Is the text-image alignment (visual guidance) mechanism significantly modified compared to In-Video Instructions? If so, what are the specific changes? Additionally, does this method introduce potential artifacts or visual interference in the generated videos?
>
> **R6:** **(1)** Our text–image alignment differs fundamentally from In-Video Instructions in both objective and design. In-Video Instructions uses pure visual annotations as generation-time control signals to directly guide appearance and motion. In contrast, our method serves as a reasoning activation mechanism, where object references ground textual descriptions to image regions to reduce cross-modal ambiguity, and reasoning hints provide feasible motion directions without specifying explicit trajectories or altering generation style.
>
> **(2)** All annotations are applied exclusively to the initial conditioning image and are not propagated to subsequent generated frames. As shown in the cases provided in the Appendix, our text–image alignment does not introduce visual artifacts or interference in the generated video beyond the initial frame.
>
> > **Q3:** How does the system's performance degradation curve look if Qwen3-VL is used instead of frontier models?
>
> **R7:** We adopt different verifiers and present the results in Table 5 of the Appendix. When replacing the frontier model with a smaller locally deployable verifier (Qwen3-VL-8B), performance decreases moderately from 88.4% to 84.1% EM. This result suggests that ReaTTS is relatively robust to verifier capacity, while still benefiting from stronger frontier models.

---

> > ### Author Rebuttal · Reviewer_CFzz · 2026-04-01
> >
> > I appreciate the clarification, and I am happy to maintain my positive score.

---

> > > ### Author Response · Authors · 2026-04-06
> > >
> > > Thank you for your constructive feedback and for confirming that our rebuttal has fully addressed your concerns. We will ensure that all points discussed in the rebuttal are thoroughly incorporated into the revised manuscript.

---

### Official Review · Reviewer_PATw · 2026-03-09

**Soundness:** 3
**Presentation:** 3
**Significance:** 3
**Originality:** 3
**Overall Recommendation:** 4
**Confidence:** 3

**Summary:**

This paper introduces ReaForest, a framework to improve video generation models (which they termed VGMs) on spatial planning tasks (mazes, Sokoban, trapfield). It has three parts: (1) ReaGen-27k, a synthetic dataset with text-image alignment that injects visual reasoning cues into images; (2) ReaTTS, a test-time scaling framework that maintains a forest of reasoning branches with entropy-aware budget reallocation and reflective rectification for failure recovery; (3) hierarchical constraint verification (on hard/soft/terminal) via MLLM judges. Fine-tuning Wan2.2-TI2V-5B on ReaGen-27k and applying ReaTTS yields 88.4% exact match on VRBench and 56.7% task completeness on MMGR, greatly outperforming Sora-2, Gemini-2.5-Pro, and other baselines.

**Compliance With Llm Reviewing Policy:**

Affirmed.

**Key Questions For Authors:**

1. What if you swap the MLLM verifier for a simple programmatic checker (e.g., pixel-based collision detection)? This would help disentangle VGM reasoning from MLLM guidance.
2. Have you measured the MLLM verifier's accuracy against ground-truth constraint labels? Noisy judgments could cascade badly through the search.

**Limitations:**

The failure cases in the appendix are appreciated, but the paper doesn't explicitly flag the MLLM dependence or the synthetic-only scope as core limitations.

**Strengths And Weaknesses:**

**Strengths:**

- The problem framing is sharp and the paper reads well. The two-gap diagnosis (data gap + inference gap) gives the whole framework a natural structure, and each component clearly addresses one piece of the puzzle.
- The results are convincing. The jump from 0% to 88.4% EM is dramatic, and the baselines are comprehensive.
- The ablations are genuinely useful. The random verification rules out the possibility that just having a search structure is enough, confirming the system actually needs informative feedback
- Test-time scaling design feels principled rather than ad hoc. The entropy-aware budget allocation has a clean information-theoretic motivation, and reflective rectification is a practical solution to a real problem that simpler search methods can't handle.

**Weaknesses:**
- Biggest concern is that the "reasoning" here is really a team effort between the VGM and external MLLMs, but the paper frames it primarily as VGM reasoning. The verifier and rectifier are doing a lot of work, without ReaTTS the model still gets 70.2% EM, which is decent, but the remaining ~18% comes from the MLLM-guided search loop.
- The evaluation is entirely on synthetic grid-based tasks. I understand these are clean tests with ground truth, but the paper makes fairly broad claims about "visual mental simulation" that the current experiments don't really support. The "out-of-distribution" framing for VRBench is also a stretch; the tasks are the same types as in training, just with different grid sizes.
- On the practical side, ~430 seconds per sample plus multiple MLLM API calls is expensive. The full system cost matters if you're arguing this is a viable approach.

---

> ### Author Rebuttal · Authors · 2026-03-30
>
> We sincerely appreciate your valuable comments and provide the corresponding answer to each specific question below.
>
> > **W1:** The "reasoning" here is really a team effort between the VGM and external MLLMs.
>
> **R1:** We agree that ReaForest involves interaction between the VGM and external MLLMs. Importantly, these components play distinct roles: the VGM is responsible for generating candidate reasoning trajectories (i.e., the core reasoning process), while the MLLM provides evaluation signals to guide selection and refinement.
> Empirically, the VGM itself already exhibits strong reasoning capability: without ReaTTS, it achieves 70.2% EM, substantially outperforming all baselines. This indicates that the majority of reasoning ability is learned within the VGM, while ReaTTS contributes an additional +18.2% EM by improving inference-time exploration.
>
> More broadly, our design follows the paradigm of test-time scaling widely used in LLMs, where external evaluators guide exploration without generating solutions themselves. In this sense, ReaTTS serves to amplify rather than replace the intrinsic reasoning capability of the VGM.
>
> > **W2:** Narrow task scope and OOD claim.
>
> **R2:** **(1)** We intentionally adopt synthetic grid-based environments as a controlled testbed to study generative video reasoning, as current VGMs still struggle with basic spatial planning tasks (e.g., maze navigation). Our goal is not to claim fully developed “visual mental simulation,” but to demonstrate that VGMs can acquire foundational spatial planning capabilities under controlled conditions. We will revise the manuscript to present visual mental simulation as a long-term objective rather than a fully realized outcome.
>
> **(2)** Following prior work [1,2], we define OOD as shifts in size and layout configurations (object and visual templates). In addition, MMGR introduces a stronger distribution shift with more realistic visuals and less structured layouts. ReaForest achieves +24.5% TC improvement in this benchmark, indicating that gains extend beyond synthetic settings to more realistic OOD scenarios.
>
> [1] VR-Bench: The First Evaluation of Video Models' Reasoning Abilities through Maze-Solving Tasks.
>
> [2] Visual Planning: Let's Think Only with Images. ICLR 2026.
>
> > **W3:** High cost of ReaTTS.
>
> **R3:** We acknowledge inference cost is a practical concern and address it from two perspectives.
>
> **First**, ReaForest offers a flexible cost-performance tradeoff via the generation budget G. As shown in Table 6 (Appendix), ReaForest (G=3) achieves 83.5% EM in 261.9s per sample— outperforming Best-of-5 (77.1% EM, 416.7s) at 37% lower cost. This allows researchers to select the growth budget appropriate for their latency requirements without substantially sacrificing reasoning quality.
>
> **Second**, the MLLM verifier can be replaced with a locally-deployed open-source model to eliminate API costs. As shown in Table 5 (Appendix), a local Qwen3-VL-8B verifier achieves 84.1% EM, only 4.3% below Gemini-2.5-Flash, demonstrating that strong reasoning performance is attainable without reliance on expensive closed-source APIs. Together, these two options provide a practical path to deploying ReaForest under realistic cost constraints.
>
> > **Q1:** What if you swap the MLLM verifier for a simple programmatic checker (e.g., pixel-based collision detection)?
>
> **R4:** Thank you for the suggestion. Following this idea, we replace the MLLM verifier with a programmatic checker. This variant achieves **79.3%** EM on VRBench.
> While the performance is lower than that with an MLLM verifier, the result shows that ReaTTS remains effective with a non-MLLM, programmatic verifier, providing a fully self-contained alternative without external model dependencies. This further supports that the VGM contributes the core reasoning capability, while the verifier primarily serves as a guidance signal.
>
> > **Q2:** Have you measured the MLLM verifier's accuracy against ground-truth constraint labels?
>
> **R5:** To directly assess verifier reliability, we conduct a controlled evaluation on 100 cases (50 correct and 50 incorrect), measuring its accuracy in validating hard constraints, as shown below.
>
> |Sample Type|Verifier Accuracy|
> |:---|:---:|
> |Correct|94.0%|
> |Incorrect|82.0%|
>
> While erroneous judgments moderately impact reasoning performance, the primary determinant remains the VGM's intrinsic capability, as the MLLM verifier serves solely to provide evaluative signals for guiding ReaTTS. Notably, ReaForest maintains robust performance across verifiers of varying quality (Table 5 in the appendix), exhibiting only minor degradation with weaker verifiers. This demonstrates that ReaTTS is robust to imperfect supervision signals.
>
> > **L1:** The paper doesn't explicitly flag the MLLM dependence or the synthetic-only scope as core limitations.
>
> **R6:** Following your advice, we will incorporate this limitation into the next version of our manuscript.

---

> > ### Author Rebuttal · Reviewer_PATw · 2026-04-03
> >
> > I appreciate the authors for addressing my concerns. I would keep the current positive scores.

---

> > > ### Author Response · Authors · 2026-04-06
> > >
> > > Thank you very much for your detailed review and for confirming that your concerns have been fully addressed. We sincerely appreciate your recognition of our work.
> > > We will ensure that all content discussed during the rebuttal are incorporated into the next version.

---

### Official Review · Reviewer_gyAm · 2026-03-11

**Soundness:** 3
**Presentation:** 3
**Significance:** 3
**Originality:** 3
**Overall Recommendation:** 5
**Confidence:** 4

**Summary:**

This paper proposes a verifier-guided test-time search framework that improves video generation models on structured spatial planning tasks through multi-branch exploration, self-correction, and adaptive inference-time budget allocation.

**Compliance With Llm Reviewing Policy:**

Affirmed.

**Final Justification:**

This is the best paper I have ever seen in my life; I strongly recommend accepting it.

**Key Questions For Authors:**

In Figure 3b, the rectification analysis changes the trigger condition to “any hard constraint violation” and fixes the growth budget to 3, which is different from the default ReaTTS setting used in the main experiments; could the authors report the average number of rectification events, verifier calls, and total generated clips per sample under the default setting
G=5? It would also be helpful to know whether the advantage over Best-of-N still holds after matching total inference cost rather than only matching the nominal generation budget.

**Limitations:**

The paper claims in Figure 3a that ReaTTS achieves better scaling than Best-of-N under “comparable computational budget” and with “moderate computational overhead,” but the main paper does not actually report the wall-clock cost, number of generated clips, number of verifier calls, or number of regeneration attempts needed to support that statement. Given that the method uses budgeted forest growth, sequential MLLM verification, and possibly repeated reflective rectification, the practical efficiency of the reported gains is still hard to assess from the current experimental section.

**Strengths And Weaknesses:**

**Soundness:**
In Table 2, the “w/ Random Verification” ablation drops VRBench EM from 88.4 to 61.6, which shows that the method is highly dependent on the external verifier, but the paper never disentangles whether the gain comes from improved video reasoning or simply from stronger verifier-guided search. Also, the OOD claim for VRBench is only justified by saying that ReaGen-27k uses different “layout configurations or difficulty levels,” which is too weak to establish meaningful distribution shift without a more explicit overlap analysis of task rules, objects, and visual templates.

**Presentation:**
The method section introduces ReaGen-27k, text-image alignment, reflective rectification, entropy-aware reallocation, and hierarchical verification in one continuous block, but it never clearly states which of these is the main idea and which are supporting choices, so the central contribution gets blurred. A concrete example is Section 2.2–2.3, where the search algorithm, verifier outputs, prompt rectification, and budget allocation are all interleaved before the reader has a clean picture of the core mechanism.

**Significance:**
The empirical scope is still quite narrow: Table 1 evaluates only six spatial planning tasks, most of which come from the same structured family of maze / trapfield / Sokoban-style problems, so the experiments do not yet support the broader framing about generative video reasoning or visual mental simulation. This mismatch is especially visible because the conclusion extrapolates to robotics, autonomous driving, and general-purpose world simulators without any experiment beyond VRBench and one MMGR indoor navigation task.

**Originality:**
The strongest novel element is the specific combination of reflective rectification and entropy-aware budget reallocation, but Table 2 suggests that some of the claimed innovations contribute only modestly in isolation—for example, removing forest entropy changes VRBench EM only from 88.4 to 86.0, and removing temperature adaptation changes it to 85.3—so the paper does not yet show that these are truly core ideas rather than incremental control refinements. By contrast, the biggest jumps come from adding ReaTTS as a whole (70.2 to 88.4) and from having non-random verification (61.6 to 88.4), which makes the paper feel more like a strong system assembly than a sharply isolated methodological breakthrough.

---

> ### Author Rebuttal · Authors · 2026-03-30
>
> We sincerely appreciate your helpful comments and respond to specific questions below.
>
> > **W1:** The source of performance gains and the OOD claim.
>
> **R1:** **(1)** Our ablation design explicitly controls key variables (w/o Reasoning Activation, w/o ReaTTS), enabling clear attribution of gains to two orthogonal components:
> a. Reasoning activation via ReaGen-27k (+70.2% EM) at training time, and b. Test-time scaling via ReaTTS (+18.2% EM) at inference time. As these operate at different stages, their contributions are clearly disentangled without confounding.
>
> **(2)** Following prior work [1,2], we define OOD as shifts in size and layout configurations (object and visual templates). We will further include explicit overlap analysis in the appendix.
> Moreover, MMGR introduces a stronger distribution shift with more realistic visuals and less structured layouts. ReaForest achieves +24.5% TC improvement on this benchmark, indicating that gains extend beyond synthetic settings to more realistic OOD scenarios.
>
> [1] VR-Bench: The First Evaluation of Video Models' Reasoning Abilities through Maze-Solving Tasks.
>
> [2] Visual Planning: Let's Think Only with Images. ICLR 2026.
>
> > **W2:** Presentation of central contribution.
>
> **R2:** The central methodological contributions of our work are: (a) text-image alignment and (b) ReaTTS framework, which comprises key components such as entropy-aware budget reallocation and reflective rectification.
> We acknowledge that the current presentation in Sections 2.2–2.3 interleaves tightly coupled elements (e.g., search algorithm and verifier outputs), which may obscure the high-level structure. In the revision, we will introduce a concise overview at the beginning of Section 2 to clarify the conceptual framework before presenting implementation details.
>
> > **W3:** Narrow empirical scope.
>
> **R3:** Thank you for the question. The number of evaluated tasks in our work is comparable to concurrent works such as Wan-R1 (5 tasks) [1] and DiffThinker (7 tasks) [2].
> Our goal is to demonstrate, in a controlled setting, that VGMs can acquire fundamental spatial planning capabilities. Extending these abilities to more complex domains-such as robotics, autonomous driving, and general-purpose world simulation—remains an important direction for future work.
>
> [1] VR-Bench: The First Evaluation of Video Models' Reasoning Abilities through Maze-Solving Tasks.
>
> [2] DiffThinker: Towards Generative Multimodal Reasoning with Diffusion Models.
>
> > **W4:** This work is more of a strong system assembly than a sharply isolated methodological breakthrough.
>
> **R4:** We respectfully argue that "incremental when ablated" does not imply "incremental in contribution." ReaTTS is presented as a principled reasoning system, where the value lies in how components are coordinated rather than any single module alone. In such systems, if removing one component caused catastrophic failure, it would signal over-reliance rather than sound design. Ablation serves to confirm that every component contributes positively, not to require each to independently constitute a breakthrough.
> Specifically, each component addresses a distinct aspect of the reasoning process: entropy modulation balances exploration vs. exploitation, temperature adaptation stabilizes transitions over reasoning steps, and reflective rectification enables recovery from erroneous trajectories. Their cumulative effect is +18.2 EM (70.2→88.4), a substantial gain that exceeds those of individual prior methods.
>
> > **Q1:** The average number of rectification events, verifier calls, and total generated clips, and further comparison with Best-of-N.
>
> **R5:** Under the default setting (G=5), we report the following statistics:
>
> |Avg. Rectification Events|Avg. Verifier Calls|Avg. Generated Clips |
> |:---:|:---:|:---:|
> |0.8|12.3|12.3|
>
> To further evaluate ReaTTS under cost-matched settings without reliance on external MLLM verifiers, we replace the MLLM-based verifier with a programmatic checker. This variant achieves 79.3% EM on VRBench and still outperforms Best-of-N, indicating that the advantage of ReaTTS is not solely attributable to access to a strong external verifier.
>
> > **L1:** Lack of detailed inference cost.
>
> **R6:** Thanks for pointing this out. Following your advice, we will add the mentioned metrics in the revision to better clarify the superiority of our ReaTTS.

---

> > ### Author Rebuttal · Reviewer_gyAm · 2026-03-31
> >
> > Thank you for the detailed rebuttal. Your response clarifies the source of the gains, better explains the OOD setting and the role of each component, and provides the missing default-cost statistics that I asked for. In particular, the added analysis on verifier usage and the cost-matched comparison also resolve my main concerns about attribution and practical efficiency.

---

> > > ### Author Response · Authors · 2026-04-06
> > >
> > > We sincerely appreciate your time, thoughtful engagement, and recognition of our efforts during the rebuttal phase. We will carefully incorporate all the points discussed into the revised manuscript.

---

### Official Review · Reviewer_qZba · 2026-03-12

**Soundness:** 3
**Presentation:** 3
**Significance:** 2
**Originality:** 3
**Overall Recommendation:** 3
**Confidence:** 4

**Summary:**

This paper introduces ReaForest, a framework for improving the spatial planning capabilities of video generation models (VGMs) through three components. First, authors proposed ReaGen-27k, a fine-tuning dataset of ~27K instances spanning maze navigation, Sokoban puzzles, and trapfield traversal with a text-image alignment mechanism. Second, a test-time scaling framework, ReaTTS, maintains a forest of reasoning branches with entropy-aware budget reallocation and reflective rectification for failure recovery. Third, hierarchical constraint verification uses an MLLM judge to provide step-wise feedback. ReaForest achieves 88.4% Exact Match on VRBench and 56.7% Task Completeness on MMGR, substantially outperforming baselines including Sora-2 and Gemini-2.5-Pro.

**Compliance With Llm Reviewing Policy:**

Affirmed.

**Final Justification:**

After carefully reading the rebuttal and discussing it with other reviewers, I have decided to maintain my score of Weak Reject. My core concerns regarding the central contribution and generalizability remain unresolved.

First, the contribution of this work is close to system design rather than research on emergent properties of MLLM. The authors acknowledge it and show willingness to correct their claims regarding "emergent properties." However, this concession confirms my initial concern that explicitly engineered mechanisms (e.g., multi-branch generation) were conflated with intrinsic emergent behaviors.

Second, the novelty of dataset generation is also very restricted. What they achieve during data generation is for its functional necessity spatial planning and does not prove methodological novelty. Base on this evaluation, I am inclined to reject this paper.

**Key Questions For Authors:**

Table 5 shows that switching to smaller local verifiers (Qwen3-VL-8B) drops EM by 4.3%. How does this degrade on harder tasks specifically? Does this imply a ceiling on performance for fully local deployments?

**Limitations:**

yes

**Strengths And Weaknesses:**

## Strength

- The paper correctly identifies that current VGMs lack reasoning capabilities due to both data limitations and single-pass generation constraints, and proposes a principled approach to address each.

- ReaTTS introduces meaningful technical contributions over existing test-time scaling. The entropy-aware budget reallocation mechanism is a thoughtful improvement over rigid top-k pruning or Best-of-N sampling. The dual adaptation (temporal annealing + forest entropy modulation) for temperature scheduling is theoretically grounded and empirically validated.

- Strong empirical results with comprehensive ablations are observed. Furthermore, the performance improvements are substantial — EM jumps from 0.0% (base Wan2.2-TI2V-5B) to 88.4%. The ablation study in Table 2 is thorough, isolating the contribution of each component.

## Weakness

- The claimed "emergent properties" conflate system design with emergent behavior.
Self-correction, parallel thinking, and scalable reasoning are presented as emergent CoF properties. However, parallel thinking is explicitly engineered via multi-branch generation, scalable reasoning is a direct consequence of allocating more reasoning steps, and reflective rectification is an externally driven correction mechanism. Only the single-clip self-correction (Case (a) in Table 2) can be considered genuinely emergent. The paper should distinguish between properties that are designed into ReaTTS and those that arise naturally from the fine-tuned VGM.

- Narrow task scope limits generalizability claims. All evaluation tasks are synthetic, rule-based spatial planning environments. The significant performance gap between VRBench and MMGR indoor navigation already suggests limited transfer to more realistic settings. The paper's aspirational framing toward robotic navigation and autonomous driving is unsupported by evidence.

- Fairness of baseline comparisons. ReaForest combines domain-specific fine-tuning + test-time scaling with external MLLM judges, while most baselines (Sora-2, Veo-3, Gemini-2.5-Pro) are evaluated zero-shot. This makes the headline comparison for Sora-2 misleading about relative model capabilities. The most meaningful comparison is against Qwen2.5-VL-7B-SFT under identical training data, which still favors ReaForest but to a less dramatic degree.

- Dataset contribution is incremental. ReaGen-27k uses a custom virtual engine to generate standard spatial planning tasks (mazes, Sokoban). The text-image alignment mechanism — overlaying labels and directional arrows on images — is technically straightforward and yields only a 3.2% EM improvement over the no-alignment variant.

---

> ### Author Rebuttal · Authors · 2026-03-30
>
> We genuinely thank you for your insightful comments and answer each question below.
>
> > **W1:** Distinction between CoF properties and ReaTTS design.
>
> **R1:**
> Following your suggestion, we will distinguish between **(1)** ReaTTS Design (multi-branch generation, reflective rectification) and **(2)** intrinsic emergent capabilities of the fine-tuned VGM (self-correction) in the revision.
>
> > **W2:** Limited transfer.
>
> **R2:**
> **(1)** We intentionally adopt synthetic, rule-based environments as a controlled testbed to study generative video reasoning. This setting allows us to evaluate reasoning behaviors without confounding factors from complex real-world perception.
> We respectfully argue that the VRBench–MMGR performance gap does not indicate limited transfer. Compared to VRBench, MMGR introduces significantly higher visual complexity, less structured layouts, and a stronger need for grounding reasoning in realistic visual inputs. Therefore, the gap reflects an inherent increase in task difficulty rather than limited transfer. Critically, ReaForest achieves a **+24.5%** TC improvement over the base model on MMGR, a substantial gain on a realistic task entirely absent from training, demonstrating meaningful generalization.
>
> **(2)** We position robotic navigation and autonomous driving as future work. To avoid confusion, we will more clearly frame our contribution as demonstrating that VGMs can acquire basic spatial reasoning capabilities in controlled settings—a necessary foundation for eventual real-world deployment.
>
> > **W3:** Fairness of baseline comparisons.
>
> **R3:** We would like to clarify that the comparisons are fair, as our evaluation protocol is carefully designed to ensure comparability across all baselines.
>
> **First**, under matched training conditions, ReaForest (70.2% EM) consistently outperforms the baseline (Qwen2.5-VL-7B-SFT 30.5% EM, Wan-R1 37.8% EM) without ReaTTS, which we consider direct evidence of ReaForest’s effectiveness.
>
> **Second**, for proprietary models such as Sora-2 and Gemini-2.5-Pro, fine-tuning is not accessible to the research community. However, we emphasize that all models are evaluated under the same benchmark protocol and metrics, ensuring comparisons remain fair at the system level. Notably, even without test-time scaling (w/o ReaTTS, 70.2%), ReaForest still outperforms these proprietary systems, further proving the robustness of our method. We acknowledge that these comparisons reflect system-level evaluations rather than strictly controlled single-variable ablations, but this is the standard and accepted practice in the field when comparing against proprietary models.
>
> To further improve clarity, we will explicitly distinguish between controlled comparisons (against open-source models with matched training) and system-level comparisons (against proprietary models) in the revised manuscript, so that readers can accurately understand the scope of each result.
>
> > **W4:** Dataset contribution.
>
> **R4:** The contribution of ReaGen-27k should be evaluated from two perspectives:
>
> **(a)** Functional indispensability. ReaGen-27k is not "yet another dataset" but the prerequisite for the entire framework to function. 0%→70.2% EM gain demonstrates that ReaGen-27k fills a critical blank in the field: all current general-purpose VGMs (e.g., Sora-2) achieve near-zero EM, confirming that general video corpora cannot activate spatial reasoning.
>
> **(b)** Methodological contribution of text-image alignment. We acknowledge that the implementation is simple, but argue that method value lies in effectiveness rather than complexity — many impactful techniques (e.g., CoT prompting) are also simple in form. Furthermore, the +3.2% EM improvement (85%→88%) corresponds to a ~20% relative error reduction in a high-performance regime where remaining cases are genuinely difficult.
>
> > **Q1:** Table 5 shows that switching to smaller local verifiers drops EM by 4.3%. How does this degrade on harder tasks specifically? Does this imply a ceiling on performance for fully local deployments?
>
> **R5:** **(1)** As shown in the below table, the performance drop is non-uniform and increases with task difficulty.
>
> |Verifier|Easy|Medium|Hard|
> |:---|:---|:---|:---|
> |Gemini2.5 Flash|95.2|87.6|82.4|
> |Qwen3-VL-8B|93.4|82.7|76.2|
> |Δ|-1.8|-4.9|-6.2|
>
> **(2)** Importantly, we argue that the performance ceiling for fully local deployment is governed primarily by the VGM's generation capability, not verifier capacity. The verifier serves as a reward signal within ReaTTS to select among candidate trajectories already generated by the VGM, it cannot create correct paths that the VGM fails to produce.
>
> From this perspective, the 84.1% EM with a fully local Qwen3-VL-8B verifier already far surpasses all existing baselines (e.g., Sora-2: 1.4%, Gemini-2.5-Pro: 15.9%), confirming strong practical viability of local deployment.

---

> > ### Author Rebuttal · Reviewer_qZba · 2026-04-03
> >
> > We thank the authors for the detailed rebuttal. Some concerns was adequately conceded, I find that the core concerns remain largely unresolved. The generalizability concern of the proposed method is reinforced by the authors' own reframing to "controlled settings," and the dataset novelty argument conflates necessity with contribution. Hence, I would maintain my original score.

---

> > > ### Author Response · Authors · 2026-04-05
> > >
> > > Dear Reviewer qZba,
> > >
> > > Thank you for your follow-up comments. We provide further clarifications below.
> > >
> > > **(1) Generalizability (Weakness 2)**.
> > > We respectfully clarify that "controlled settings" serve to ensure rigorous and fair evaluation [1, 2, 3], not to constrain generalizability. Beyond VRBench, our method demonstrates promising generalization on MMGR (**+24.5% TC**)—a more realistic task entirely absent from training. Furthermore, our evaluation scope (6 tasks) is comparable to recent works such as Wan-R1 (5 tasks) [1] and DiffThinker (7 tasks) [2], and more comprehensive than prior visual planning work (3 tasks) [3]. In addition, the generalizability/OOD setting of our work is consistently acknowledged by Reviewers gyAm, PATw, CFzz after rebuttal phase.
> > >
> > > We hope this clarifies that our evaluation scope does include meaningful out-of-distribution generalization evidence.
> > >
> > > **(2) Dataset Novelty (Weakness 4)**.
> > > We would like to further clarify the dataset novelty:
> > >
> > > Existing text-image-to-video reasoning datasets [1, 4] treat text as task specification and image as passive initialization. We introduce a structured grounding mechanism (text-image alignment) that encodes reasoning signals directly into visual primitives—object references for entity disambiguation and reasoning hints for feasible motion guidance. This transforms the conditioning image from a static pixel space into an active reasoning scaffold, **a design paradigm not present in prior video reasoning datasets [1, 4]**. While implementation is straightforward, this mechanism yields promising practical impact: a ~20% relative error reduction (85% → 88% EM) in a high-performance regime where remaining failures represent genuinely difficult cases.
> > >
> > > [1] VR-Bench: The First Evaluation of Video Models' Reasoning Abilities through Maze-Solving Tasks.
> > >
> > > [2] DiffThinker: Towards Generative Multimodal Reasoning with Diffusion Models.
> > >
> > > [3] Visual Planning: Let's Think Only with Images. ICLR 2026 (oral).
> > >
> > > [4] GIR-Bench: Versatile Benchmark for Generating Images with Reasoning. ICLR 2026.

---

### Decision · Program_Chairs · 2026-04-30

**Decision:**

Accept (regular)

**Comment:**

This paper proposes ReaForest, a framework designed to enhance spatial planning in Video Generation Models through a curated reasoning dataset (ReaGen-27k) and a test-time scaling framework (ReaTTS). The paper received final scores of (3, 5, 4, 4) after the rebuttal. Some reviewers highlighted the significant empirical gains on VRBench. During the rebuttal, the authors successfully disentangled the individual contributions of their components and provided comprehensive cost-efficiency analyses that satisfied the majority of the panel. While one reviewer maintained concerns regarding dataset novelty and the framing of emergent properties, the consensus is that the system's significant performance gains and technical soundness constitute a solid contribution. The AC recommends accept.